# What does the Knowledge Neuron Thesis Have to do with Knowledge?

**Jingcheng Niu**[14]  **Andrew Liu**[2]  **Zining Zhu**[134]  **Gerald Penn**[14]
niu@cs.toronto.edu  a254liu@uwaterloo.ca  zzhu41@stevens.edu  gpenn@cs.toronto.edu
[1]University of Toronto, [2]University of Waterloo, [3]Stevens Institute of Technology, [4]Vector Institute

## Abstract

We reassess the Knowledge Neuron (KN) Thesis: an interpretation of the mechanism underlying the ability of large language models to recall facts from a training corpus. This nascent thesis proposes that facts are recalled from the training corpus through the MLP weights in a manner resembling key-value memory, implying in effect that "knowledge" is stored in the network. Furthermore, by modifying the MLP modules, one can control the language model's generation of factual information. The plausibility of the KN thesis has been demonstrated by the success of KN-inspired model editing methods (Dai et al., 2022; Meng et al., 2022).

We find that this thesis is, at best, an oversimplification. Not only have we found that we can edit the expression of certain linguistic phenomena using the same model editing methods but, through a more comprehensive evaluation, we have found that the KN thesis does not adequately explain the process of factual expression. While it is possible to argue that the MLP weights store complex patterns that are interpretable both syntactically and semantically, these patterns do not constitute "knowledge." To gain a more comprehensive understanding of the knowledge representation process, we must look beyond the MLP weights and explore recent models' complex layer structures and attention mechanisms.

## 1 Introduction

Recent research has highlighted the remarkable ability of large pretrained language models (PLMs) to recall facts from a training corpus (Petroni et al., 2019). The underlying mechanism by which this information is stored and retrieved within PLMs, however, remains a subject of intensive investigation. The Knowledge Neuron (KN) Thesis has been recently proposed as a novel framework for interpreting language models (LMs) (Dai et al., 2022; Meng et al., 2022; 2023). This thesis suggests that LMs operate akin to key-value memories, recalling facts from the training corpus through the multi-layer perceptron (MLP) weights. Therefore, a significant implication of the KN thesis is that factual information generation by LMs can be controlled by modifying the MLP modules. Should this manipulation of factual information recall become feasible, it could lead to the development of language models that are more controllable, interpretable, and factually aligned.

The plausibility of the KN thesis is demonstrated by the success of KN-inspired model-editing methods. Dai et al. (2022) argued that relational facts can be localised to a handful of 2-5 MLP neurons. They then developed a method to identify these neurons using a search algorithm based on an integral of gradients. By manipulating the activation of these identified neurons (*KN edit*), they managed to alter the model's response to fill-in-the-blank cloze tasks and generate counterfactual information without additional fine-tuning. In a parallel approach, Meng et al. (2022) proposed a more intricate model wherein factual recall occurs in two critical locations, each incorporating a different module. In this model, the mid-layer MLP retrieves the fact, and an attention module copies it into the output response at the topmost layer. Despite this proposed two-step process, their proposed model editing method, Rank-One Model Editing (ROME), only modifies MLP weights, much as KN edit only modifies MLP activations without editing attention modules.

While the efficacy of these model editing methods has been showcased in simple fill-in-the-blank cloze tasks, the appraisal of such achievements mainly rests on basic paraphrasing of the prompts, as

Figure 1: Syntactic phenomena can be located and edited using existing model editing methods. The integrated gradient of singular determiner (*this*, *that*) and plural determiner (*these*, *those*) form two distinct groups. Erasing these neurons leads to output probability changes.

outlined by Yao et al. (2023), who introduced an additional assessment metric, *portability*, finding that model-editing methods to date lack robustness. Their performance is halved when evaluated with the portability measure. Building on this, we introduce two new metrics. First, a successful edit must demonstrate symmetry within bijective relationships (e.g., with the assertion *Ottawa is the capital of Canada*, the reciprocal *Canada's capital is Ottawa* should also hold valid). Second, a successful edit must extend to synonym usage (e.g., *a dentist treats a toothache* and *a dentist treats tooth pain* should be considered equivalent). Our evaluation shows that existing model-editing methods are even less robust under these two new criteria.

It is practically impossible to exhaustively assess factual model-editing methods due to the difficulty in systematically dealing with counterfactual data. The potential counterfactual replacements for Canada's capital are seemingly endless. Thus, beyond the introduction of the two new evaluation criteria above, we propose the evaluation of model-editing methods using syntactic constructions. We have determined that the KN thesis applies just as reliably to syntactic phenomena (as illustrated in Figure 1). Unlike many facts, syntactic phenomena can provide rigorously defined targets for editing through the use of so-called minimal pairs. As a result, in this paper, we re-evaluate the KN thesis by expanding the scope of our assessment to include more complex factual patterns and syntactic phenomena. This also speaks to a long-standing debate regarding the formal *vs.* functional competence of language models (Mahowald et al., 2023) — an LM's ability to follow linguistic rules and patterns *vs.* its ability to apply language in the real-world (see §2.3). If we edit a model's expression of facts and linguistic phenomena using the same approach, this could indicate that both the formal and functional competencies of an LM are governed by the same underlying mechanisms.

Within the context of Dai et al.'s (2022) KN framework, KN edit's efficacy is unsatisfactory. Editing the KN activations has only limited impact on categorical predictions. The effect of KN edit is only apparent in the shifts in the output probability distributions of tokens. The patterns that the method localises also appeal to superficial cues such as word co-occurrence frequencies. We also find several critical shortcomings in the ROME framework. LMs process both linguistic and factual information in phases, but the exact task distribution between the MLP and attention modules appears to be more idiosyncratic than initially theorized (Meng et al., 2022). ROME model editing only superficially alters token association patterns, in a manner that is inconsistent across the various expressions that may attend the same underlying knowledge. As a result, whatever is being manipulated reflects none of the traditional tautologies that have been associated with "knowledge," as that term has been understood in philosophy since the time of Aristotle. When implemented on syntactic constructions, furthermore, the influence of ROME's editing is limited only to the word altered and no pivot that preserves any reasonable standard of syntactic paraphrase, such as substitutability *salva veritate,* is forthcoming. Furthermore, ROME fails under our newly proposed symmetry and synonymy criteria.

We therefore argue for the position that the feed-forward MLP modules of the transformer model do not store knowledge, but rather complex "token expression patterns." These token expression patterns can often be interpreted linguistically, but the information that they express does not fit into linguistically or factually defined categories. A key-value, memory-based view of the language model is overly simplistic in explaining the remarkable ability of recent PLM's formal, and perhaps even functional, competence. We need to investigate the rich layer and attentive structure of recent PLMs more to arrive at a better understanding of their underlying mechanics.

In the following sections, we will first provide an overview of the KN thesis (§2). Then we will evaluate two practices inspired by it: Dai et al.'s (2022) KN edit framework (§3) and Meng et al.'s (2022) ROME framework (§4). Finally, we will conclude the paper with a discussion (§5).[1]

---

[1]The code, data and results are publicly available at https://github.com/frankniujc/kn_thesis.

## 2 THE KNOWLEDGE NEURON THESIS

Geva et al. (2021) were among the first to propose that the MLP modules in a transformer model behave like key-value memories. A typical MLP module in recent transformer-based PLMs has two layers. They argue that the first layer corresponds to keys, and the second layer, to values.[2] They found that each key neuron is triggered by human-interpretable shallow input patterns such as periods of time that end with the letter "*a*." Then, the corresponding value neurons distorted the next-token output probability, until a final distribution is generated.

The KN thesis emerged as a result of this important discovery. Dai et al. (2022) coined the term *knowledge neuron* and ambitiously claimed that the keys and values within MLP modules not only capture simple patterns but also store "knowledge." They formulate an item of fact, such as *Canada's capital is Ottawa*, as a 3-tuple $(s, t, r)$, consisting of the source ($s$, *Canada*), the target ($t$, *Ottawa*) and the relation ($r$, *capital*) between them. The authors asserted that this tuple can be localized to a small group of MLP neurons typically found in the topmost layers of the language model, which they identified by analysing the magnitude of the integrals of gradients among prompts. To support their claim, they conducted model-editing experiments. By suppressing the KNs (setting their activations to zero), they observed a decrease in the probability of generating the correct original target ($t$), while other tokens remained largely unaffected, demonstrating a "minimally invasive surgery." Meng et al. (2022) proposed a refinement of Dai et al.'s (2022) model. They employed a causal mediation method (Finlayson et al., 2021) to form a more intricate version of the KN thesis. They argue that the factual association process happens at two locations: a mid-layer MLP recalls the fact from memory, and the topmost layer's attention model copies that information to the final output.

There were similar investigations of neurons prior to the KN thesis. Durrani et al. (2020) observed the neurons of an auxiliary probing model that was trained on BERT embeddings, not the neurons of BERT itself. Therefore, their analysis faced an all-too-common dilemma for probing: did they find insights about the language models or artefacts of the fine-tuning process (Hewitt & Liang, 2019)? Finlayson et al. (2021) used causal mediation analysis to study subject-verb agreement in GPT and XLNet (Yang et al., 2019). In particular, they observed a difference in ratios between the verb with the correct inflection and one with the incorrect inflection. They then modify the prompt, see the probability change and reason about the internal mechanisms of the model for expressing subject-verb agreement. They concluded that the upper-middle layers are more relevant to the expression and that there are various levels of overlap between the top 5% neurons used to express agreement. These insights, however, just as with previous probing work, are still purely observational and largely preoccupied with layers and network depth. They are able to observe many characteristics of the process, but still cannot cannot provide a satisfactory understanding of how it happens.

More recently, there has been interest in utilizing large language models (LLMs) to gain insight into the differing functionalities of individual neurons. Despite its title's strident claim that neurons in LMs can be "explained," Bills et al. (2023) clarify that their model "explains correlations, not mechanisms." From a knowledge-representation standpoint, their evaluation of LLM explanations is also entirely observational. When Huang et al. (2023) reassessed the validity of these explanations, even the most confident ones had high error rates and little to no causal effects on the interventions that use the explanations. The LLM interpretation of LMs is still immature.

### 2.1 EVALUATING THE KN THESIS: AN OVERVIEW

The effectiveness of a model-editing algorithm is customarily evaluated across three dimensions (Yao et al., 2023): (1) **reliability**: whether the model can successfully change its output from $t$ to $t^*$ (also referred to as an *efficacy score* by Meng et al. (2022)); (2) **generality**: whether the effect is applicable to rephrased relations; and, (3) **locality**: whether the edit impacts unrelated relations. Yao et al. (2023) stress, however, that the assessment of generality is often constrained to simple paraphrasing. This is typically done by developing multiple templates for a specific relation. For instance, the relation *capital* can be structured as both "The capital of [s] is [t]." and "[s]'s capital is [t]." Previous evaluations (Elazar et al., 2021; Meng et al., 2022; 2023) prematurely announced success when a model, edited on a first template, could be generalized to a second template. Thus, Yao et al. (2023) recommended extending the assessment parameters by introducing the concept of

---

[2]Despite the similarities in nomenclature, this is unrelated to the key and value of self-attention.

*portability*. For example, having changed Watts Humphrey's *alma mater* from Trinity College to Harvard University, the model should return Boston instead of Dublin when asked about the city where Watts Humphrey received his university education. It was apparent that model-editing methods present a markedly lower level of portability than generality (50% versus 90%). The evaluation of portability, on the other hand, requires new data annotation, which can be costly.

Extending Yao et al. (2023), we attempt a more comprehensive evaluation of model editing of factual association with two extra criteria: bijective symmetry and synonymous invariance. Bijective symmetry does not require new data collection and we can obtain data automatically from previous corpora. For a bijection relation such as *capital* or *capital of*, we should see the model generalise $(s, t \rightarrow t^*, r)$ to $(t^*, s, r^{-1})$. For example, if we change the capital of Canada to Rome, then the model should also agree that Rome is the capital of Canada. Similarly, an effective edit should also be able to generalise across synonyms. If the model knows that a dentist treats toothaches, it should also know that they also treat tooth pain. Prior work (Elazar et al., 2021) only used synonym replacement on rephrasing the relation prompts — we extend it to the source and the target.

Several others have already questioned the validity of the KN thesis. Hase et al. (2023) identified discrepancies between the results of causal tracing and the effects of ROME editing. They concluded that a mechanistic understanding reveals insights on the consequences of model editing. To the best of our knowledge, we are the first to comprehensively evaluate the KN thesis using rigorously defined syntactic phenomena. We consider three: determiner-noun agreement, subject-verb agreement, and gender and number agreement across anaphoric chains.

## 2.2 Evaluating the KN Thesis on Syntactic Phenomena

Edit pairs for syntactic phenomena, by contrast, can be systematically extracted through the formation of "minimal pairs." For a grammatical sentence that expresses a linguistic phenomenon, we can construct an ungrammatical sentence that minimally differs from the original sentence in respect of one feature of grammatical acceptability. For example, the phrase *this student* can be changed to the ungrammatical counterpart, *\*this students*. The BLiMP corpus (Warstadt et al., 2020) is one of the most comprehensive and extensively utilised collections of such minimal pairs.

We therefore propose to systematically evaluate the effect of model-editing methods using syntactically differentiated prompts. We define a similar 3-tuple $(s, t, p)$ that contains the source ($s$), the target ($t$) and the syntactic phenomenon ($p$). Take the phenomenon determiner-noun agreement as an example. In a grammatical sample sentence from a minimal pair, $s$ is the tokens that are condition the expression of the target (the determiner), and $t$ is the tokens that differ within the pair (the noun). The ungrammatical target $t^*$, is the noun in the opposite form. We then intervene with model editing, and observe whether the model assigns a higher probability to $t$ than $t^*$.

## 2.3 Editing Syntactic Phenomena & the "Formal vs Functional" Distinction

If we can successfully edit facts as well as syntactic phenomena using the same model-editing methods to the same degree, then it stands to reason that the model follows a unified underlying mechanism for both factual and syntactic information. Choosing the correct city (*the Space Needle is in Seattle/\*Rome*) would be no different than choosing the correct verb form (*the apple is/\*are red*).

Mahowald et al. (2023) refers to a distinction between the formal and functional competence of a language model: formal means "knowledge of linguistic rules and patterns," and functional refers to "understanding and using language in the world." Syntactic phenomena pertain to formal competence, and facts pertain to functional competence, respectively. NLP researchers sometimes informally use the terms syntax and semantics to refer to this distinction. BLiMP even refers to anaphoric gender agreement as morphological. Jawahar et al. (2019) and Tenney et al. (2019) believe that syntactic information is located in lower layers in BERT than semantic information, because syntactic information is more "shallow." Dai et al. (2022) appear to agree with this assertion in claiming that factual information is located in the upper layers. Meng et al. (2022), however, claim that factual information is located in the middle. This contradiction may support Niu et al.'s (2022) assertion that layers are not the best explanatory device of the distribution of these types of information in LMs. We explore here the possibility that no dividing line exists at all between the mechanisms through which a language model processes information related to these two types of competence.

## 3 LOCALISING SYNTACTIC PHENOMENA IN LANGUAGE MODELS

We put the KN thesis to the test under the KN-edit framework by asking three questions: (1) can we localise linguistic phenomena using the same KN-edit method; (2) how do the levels of localisation compare to each other; and (3) are these localisations strong enough to support the KN thesis?[3]

### 3.1 METHODS: SEARCHING FOR KNs OF SYNTACTIC PHENOMENA

For each prompt, we calculate an integral-of-gradient attribution score $\alpha_i^{(l)}$ for the $i$-th intermediate neuron on the $l$-th layer ($w_i^{(l)}$). Then, for a syntactic phenomenon with the source-target pair $(s, t, p)$, we find the neurons that have an attribution score greater or equal to $\pi$=20% of the maximum attribution score shared among at least $\tau$% of its prompts. We start from $\tau$=70% and adjust it by an increment or decrement of 5% until the number of neurons is within the range of $[2, 5]$.

**Neuron Attribution Score**  Given an input prompt $x$, we follow Dai et al. (2022) and use the integral of gradients to calculate the neuron attribution score:

$$\alpha_i^{(l)} = \overline{w}_i^{(l)} \int_{\gamma=0}^{1} \frac{\partial P_x(\gamma \overline{w}_i^{(l)})}{\partial w_i^{(l)}} d\gamma, \ P_x(\hat{w}_i^{(l)}) = p(y|x, w_i^{(l)} = \hat{w}_i^{(l)}), \quad (1)$$

where $P_x(\hat{w}_i^{(l)})$ denotes the probability distribution of the token $y$ when changing the neuron $w_i^{(l)}$'s value to $\hat{w}_i^{(l)}$, and $\frac{\partial P_x(\alpha \overline{w}_i^{(l)})}{\partial w_i^{(l)}}$ denotes the gradient of the model with respect to the activation $w_i^{(l)}$. We will see a more salient gradient when the neuron inflicts a greater change on the output probability.

**Measuring the Level of Localisation**  We use three metrics to measure the level of localisation: (1) the number of identified neurons ($|KN|$) using the initial threshold setting ($\tau$=70%), (2) the final threshold $\tau$ to obtain 2-5 KNs, and, (3) a similarity score among all the token attribution patterns.

Both of Dai et al.'s (2022) measures ($|KN|$ and $\tau$) depend on adjusting the two threshold hyperparameters, $\pi$ and $\tau$. Here, we propose a nonparametric measure using a generalised $n$-sample similarity measure ($R_1^2$) that measures the correlation of all the attribution patterns:

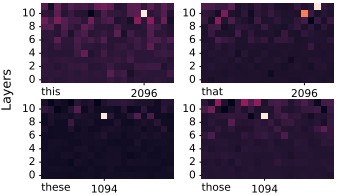

| Neuron | this | that | these | those |
|---|---|---|---|---|
| $w_{2096}^{(10)}$ | 0.93 | 0.75 | 0 | 0 |
| $w_{1094}^{(9)}$ | 0 | 0 | 1.00 | 1.00 |
| $w_{2339}^{(9)}$ | 0.33 | 0 | 0.32 | 0 |
| $w_{2686}^{(11)}$ | 0 | 0.81 | 0 | 0 |
| ... | ... | ... | ... | ... |

(a) Average KN attribution scores.  (b) KNs for Det-N pairs.

Figure 2: Localising grammatical number to KNs. The singular determiners share a common KN ($w_{2096}^{(10)}$), and the plural determiners share a different common KN ($w_{1094}^{(9)}$).

$$Y = [y_1 \ldots y_n], \ y_i = \frac{s_i}{\|s_i\|}, Y = USV^\top = \sum_{k=1}^{n} \sigma_k u_k v_k^\top, \ R^2 = \frac{\sigma_1^2 - 1}{n - 1}. \quad (2)$$

We first normalise and concatenate each attribution pattern $s_i$ for each prompt $x_i$ in the dataset into $Y$. Then, we can calculate the similarity/correlation among all $n$ patterns by conducting a singular value decomposition (SVD) and using the square of the first singular value $\sigma_1^2$. We then normalise this measure to the range $[0, 1]$ so that the similarity between $n$ parallel vectors will be $R_1^2 = 1$, and $n$ orthogonal vectors will get $R_1^2 = 0$.

### 3.2 RESULTS & FINDINGS

**Finding 1: We can localise the grammatical number of determiners to just two neurons, just like factual information.**  The BLiMP paradigm determiner_noun_agreement_2 (DNA.2) contains 1000 sentence pairs with exactly one demonstrative determiner (*this, that, these, those*) agreeing with an adjacent noun, e.g., *Carl cures those/\*that* **horses**. The determiner *those* is $t$, *that* is $t^*$ and

---

[3]Due to page restrictions, we only present the results of the determiner_noun_agreement_2 (DNA.2) paradigm on BERT in the main content. See Appendix C for the result of other BLiMP paradigms and LMs.

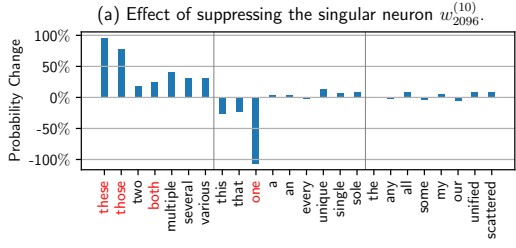 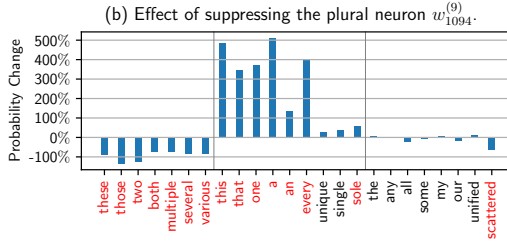

Figure 3: Suppressing the number neuron's (singular: $w_{2096}^{(10)}$; plural: $w_{1094}^{(9)}$) effect across number-expressing prenominal modifiers. Significant ($p < 0.05$) changes are highlighted in **red**. The three sections in the plots are, from left to right, plural, singular and neutral modifiers.

the noun *horses* is $s$. A noun may appear in multiple sentence pairs. Among the paradigm's 1000 sentence pairs, we identified 283 unique Det-N pairs $(s, t, t^*, r)$.

**Attribution Score Patterns** The attribution score of neurons shows a highly consistent pattern that can be interpreted linguistically. We calculated the average attribution scores of all the prompts that contains each one of the determiners. Figure 2a shows a selection of the average attribution scores. The colour block in the $i$th column and $j$th row shows the attribution score $\alpha_i^{(j)}$. As we can see, a common neuron ($w_{2096}^{(10)}$) has a high average attribution score for both of the singular determiners *this* and *that*, and another common neuron ($w_{1094}^{(9)}$) lights up for the plural determiners *these* and *those*.[4]

This pattern is not only shown in aggregate. For each Det-N pair, we use the 1000 sentences in the paradigm as templates to create the prompts needed for a KN search. For each sentence, we replace the sentence's determiner and noun with the Det-N's determiner and noun. We then obtain 1000 sentences with different contexts but the same determiners and nouns. Then, we run a KN search on these 1000 sentences. When we look into each individual Det-N pair, the two neurons are identified as KNs in the vast majority of the pairs. As shown in Figure 2b, $w_{2096}^{(10)}$ appeared in 93% of the pairs with *this* and 75% of the pairs with *that*. The plural neuron appeared in 100% of pairs with *these* or *those*. More importantly, these neurons were not identified as KNs in pairs with the opposite grammatical numbers. Figure 2b shows an excerpt of the results (full results in Appendix B.2).

**Effects of Suppressing the "Number Neuron"** Do these two neurons correspond to grammatical number? We suppress each neuron (setting activation to 0) and compute the pre- and post-edit model's output probability of various number-expressing prenominal modifiers across all prompts with singular/plural nouns. Appendix B.1 explains the prenominal modifier selection process. Figure 3 shows the average effect of suppressing the identified KNs ($\frac{p(\text{post-edit}) - p(\text{pre-edit})}{\min(p(\text{post-edit}), p(\text{pre-edit}))}$).

The result of suppressing the plural neuron is pronounced (Figure 3b). This intervention leads to a significant reduction in probability across all plural modifiers, a notable increase for the majority of singular modifiers, but a limited impact for modifiers that do not express number agreement. Therefore, erasing the activation of the plural neuron causes a decrease in the expression of determiner-noun agreement for plural modifiers. Although this KN search is solely based on these four demonstrative determiners, we observed that it generalizes to other determiners (*one, a, an, every; two, both; multiple, several, various*) and even adjectives (*single, unique, sole*). This effect is statistically significant. By treating the pre- and post-edit probabilities as two separate groups, a Student's (1908) $t$-test reveals significance when the modifiers are highlighted in **red** in

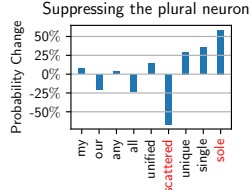

Suppressing the plural neuron.

Figure 4: The localisation of plurality appeals to word co-occurrence frequencies cues.

Figure 3. The null hypothesis is that the pre- and post-edit probabilities are sampled from the same distribution, i.e., the intervention has no effect. Thus, the neuron $w_{1094}^{(9)}$ can be interpreted through the lens of a linguistic phenomenon, viz. determiner-noun agreement.

Note, however, that the word *scattered* also sees a significant probability decrease when suppressing the plural neuron. *Scattered* does not specify for plural number; phrases such as "scattered rioting"

---

[4]We use Meng et al.'s (2022) neuron numbering system. Both layer and neuron indices start with 0.

| BLiMP Paradigm | \|KN\| | $\tau$ | $R_1^2$ | | Rels. | \|KN\| | $\tau$ | $R_1^2$ |
|---|---|---|---|---|---|---|---|---|
| det_n_agr._1 | 3.94 | 0.71 | 0.56 | | P101 | 0.167 | 0.515 | 0.399 |
| det_n_agr._2 | 1.86 | 0.62 | 0.56 | | P103 | 0.204 | 0.662 | 0.399 |
| dna._irr._1 | 5.53 | 0.73 | 0.64 | | P106 | 1.292 | 0.607 | 0.365 |
| dna._irr._2 | 2.45 | 0.67 | 0.55 | | P108 | 1.493 | 0.663 | 0.473 |
| dna._w._adj_1 | 8.88 | 0.78 | 0.67 | | P1303 | 10.462 | 0.814 | 0.684 |
| dna._w._adj_2 | 2.26 | 0.67 | 0.57 | | P140 | 2.008 | 0.689 | 0.263 |

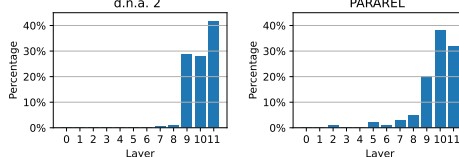

(a) Levels of localisation measures.

(b) Layer distribution of identified KNs. Both BLiMP and PARAREL occupy the topmost layers.

Figure 5: The localisation of certain syntactic phenomena (BLiMP) is comparable to facts (PARAREL). We see comparable localisation metrics and the identified KNs occupy the same layers.

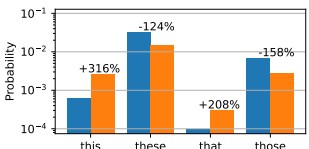

(a) The exact effect to output probability of editing the KNs. ■: pre-edit. ■: post-edit.

| Paradigm | Pre-edit | Post-edit | $\Delta$ |
|---|---|---|---|
| det_n_agr._2 | 100% | 94.8% | -5.2% |
| dna._irr._2 | 99.5% | 96.9% | -2.6% |
| dna._w._adj._2 | 97.1% | 94.4% | -2.7% |
| dna._w._adj._irr._2 | 97.4% | 95.4% | -2.0% |

(b) These modifications of determiner-noun KNs are usually not enough to overturn the categorical prediction.

| Data | Model | Reliability |
|---|---|---|
| ZsRE | T5-XL | 22.51 |
| | GPT-J | 11.34 |
| CounterFact | T5-XL | 47.86 |
| | GPT-J | 1.66 |

(c) KN edit has low reliability for facts (Yao et al., 2023).

Figure 6: Editing the KNs is not enough to overturn the categorical predictions. The major limitation of KN edit is its low reliability. These reliability scores cannot support the KN thesis.

are syntactically and semantically well-formed. But it is used more often with plural nouns because of its meaning. This frequency effect is not limited to *scattered*. Other words such as *any, all, unified*, and the three adjectives *unique, single* and *sole* exhibit a similar bias. As shown in Figure 4, we see probability changes, although less substantial, alongside those modifiers that strictly specify for grammatical number. This is a semantic number co-occurrence bias.

The suppression effect of the singular neuron is similar but less pronounced. Overall, we see the opposite effect across all prenominal modifiers, with the "singular" adjectives (*unique, single, sole*) being the only exceptions. This is, however, unsurprising. Unlike the plural neuron, the singular neuron did not appear in all of the Det-N pairs. We suspect that an LM can identify the plural property more easily when its wordpiece-based tokeniser exposes many plural suffixes.

**Finding 2: KNs obtained using linguistic tasks and factual tasks share similar characteristics of localisation.** Figure 5a shows the level of localisation of various BLiMP determiner-noun agreement paradigms and selected PARAREL relations. The localisation metrics of both BLiMP paradigms and PARAREL relations fall within the same range. See Appendix C.3 for the full list.

Furthermore, Figure 5b shows no bifurcation of layers within which linguistic and factual KNs locate (see Appendix C.2). All of the neurons are distributed in the topmost layers. The determiner-noun agreement pattern is purely syntactic. This is a refutation of Jawahar et al. (2019) and Tenney et al.'s (2019) view that syntax is localised to more shallow layers than semantics. Our results confirm Niu et al.'s (2022) assertion that the location of syntactic and semantic (and, additionally, factual) information is not distinguished by layer in the LM. In fact, our results may suggest that these types of information are most fruitfully thought of as being handled by the same functional mechanism.

**Finding 3: Despite the high level of localisation in the underlying probability drift, the effect of editing the KNs is not enough to overturn the categorical predictions made by the language model.** Although we see a high level of localisation in the relative probability change between $t$ and $t,^*$ we find that this change is often not enough to overturn the final prediction. As shown in Figure 6, we only see at most 5.2% of the BLiMP results being overturned. This low reliability issue is not limited to syntactic phenomena. In Figure 6c, we list Yao et al.'s (2023) evaluation of KN edit on two other corpora: ZsRE (Levy et al., 2017) and CounterFact (Meng et al., 2022). The reliability of the KN algorithm ranges from 1.66% to 47.86% — not enough to support the KN thesis.

**Discussion** Just as with facts, syntactic phenomena localise to neurons. Modifying merely two neurons working in tandem can significantly change the expression of determiner-noun number.

This is not the only type of localisable syntactic phenomenon (see Appendix C), and together they constitute a significant extension of Finlayson et al.'s (2021) findings — syntactic phenomena can be localised to the individual neuron level. Furthermore, these phenomena share with factual information the extent of their localisation, and the layers in which the KNs typically occur.

But do the patterns identified for these neurons constitute "knowledge?" KN edit's low reliability score and its appeal to shallow cues both suggest otherwise. If we follow the KN thesis and interpret a post-edit probability change as an indication of the quantity of knowledge stored, then we cannot draw the conclusion that knowledge is stored there. The identified neurons are spots with a high information concentration, but the final decision still lies with the rest of the model.

Interestingly, the patterns that we identified resemble linguistic categories, but they deviate from rules of grammatical well-formedness. In determiner-noun agreement, KN edit also affects pre-modifiers that do not specify for number, alongside plural-specifying determiners such as *multiple*, *several* and *various*. Phrases such as *sole breadwinners* and *scattered rioting* are less frequent but by no means unheard of. This suggests that the patterns reflected within the MLP neurons can only be completely accounted for by appealing to superficial cues such as word co-occurrence frequency.

## 4 CAUSAL TRACING AND RANK-ONE MODEL EDITING

In this section, we reassess Meng et al.'s (2022) similar but more intricate implementation of KN edit. They proposed that information is expressed at two locations: facts are recalled in mid-layer MLP weights, and copied to the final output by attention modules. They derived this thesis based on causal mediation. The causal traces in Figure 7a are computed as follows. First, the source tokens are corrupted by adding random noise $\epsilon$ and the model generates an incorrect result. Then, they restore an intermediate hidden state to its correct value for all the tokens at all layers, and determine whether this restoration can fix the corruption. They discover a division of labour between the

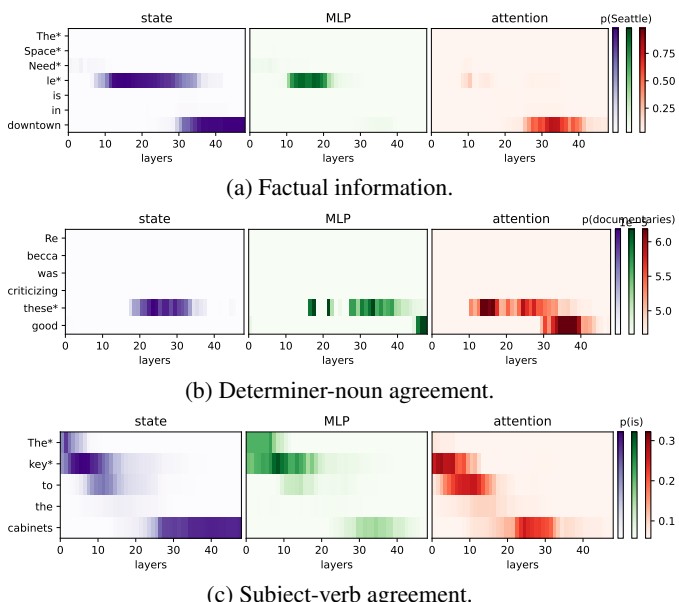

(a) Factual information.

(b) Determiner-noun agreement.

(c) Subject-verb agreement.

Figure 7: Causal tracing result.

MLP and attention. This division, however, is not stable. In Figure 7bc we reproduce this effect on syntactic phenomena. The distinction between the early and late site is no longer discernible. This is, in fact, not a distinction between facts and syntactic patterns. Many factual causal traces also do not show this distinction.[5]

Previous evaluation of the ROME model-editing method was limited to simple paraphrasing (Yao et al., 2023). We observe that ROME does not generalise well in respect of either of our new criteria, bijective symmetry or synonymous invariance (Figure 8ab). This issue persists when we evaluate ROME quantitatively. We assembled two new datasets using PARAREL relations to evaluate our two new criteria (see Appendix E for details). We use the two bijective relations R1376 and P36 to construct a bijective symmetry evaluation dataset. Then, for synonymous invariance, we rewrite the field-of-work targets in P101 into occupation names. For instance, if we change Anaxagoras's field of work from *philosophy* to *linguistics*, we also want the model to answer "Anaxagoras is a linguist" when given the prompt. Table 1 shows the result of our evaluation on these newly assembled datasets. Although ROME obtains higher reliability scores than KN edit in both GPT-2 XL and LLaMA-2 7B, the symmetry and synonymy results are both much lower. We also observe

---

[5]In Appendix D, we show the average indirect causal effects; our observation thus holds under aggregation.

| (a) **GPT-2 XL**: *The capital of Canada is Ot-tawa* 
 **ROME Edit**: Ottawa → Rome | (b) **GPT-2 XL**: *To treat my toothache, I should see a dentist* 
 **ROME Edit**: dentist → lawyer | (c) **GPT-2 XL**: *The authors near the taxi drivers* **are** 
 **ROME Edit**: are → is |
|---|---|---|
| ☺: *The capital of Canada is* **Ottawa** ... 
 ☹: *The capital of Canada is* **Rome**. | ☺: *To treat my toothache, I should see a* **dentist**, ... 
 ☹: *To treat my toothache, I should see a* **lawyer**. | ☺: *The authors near the taxi drivers* **are** ... 
 ☹: *The authors near the taxi drivers* **is** ... |
| ☺: *Ottawa is the capital of* **Canada**. 
 ☹: *Ottawa is the capital of* **Canada**'s federalist system of government. | ☺: *To treat my tooth pain, I should see a* **dentist**. 
 ☹: *To treat my tooth pain, I should see a* **dentist**. | ☺: *The authors near the dancers* in their paper **are** ... 
 ☹: *The authors near the dancers* **is** ... |
| ☺: *Rome is the capital of* **Italy**, ... 
 ☹: *Rome is the capital of* **Italy**, ... | ☺: *To treat my odontalgia, I should see a* **dentist**. 
 ☹: *To treat my odontalgia, I should see a* **dentist**. | ☺: *The pilots near the taxi drivers* **were** ... 
 ☹: *The pilots near the taxi drivers*' cabins **are** ... |
|  |  | ☺: *The pilots near the dancers* **are** ... 
 ☹: *The pilots near the dancers* **are** ... |

Figure 8: Comparison of generated text. The prompt is *italicized*, ungrammatical or counter-factual responses are highlighted in **red**, and unchanged correct responses in **green**. ☺ shows the original GPT-2 XL's generation, and ☹ shows the edited model's response.

that ROME edit can only edit the exact association between the tokens in $(s, t, r)$. As demonstrated in Figure 8c, editing the verb corresponding to *the authors* from *are* to *is* only affects the subject *the authors*, and not other subjects such as *the pilots*. These look more like at-times brittle patterns of token expression than factual knowledge.

## 5 DISCUSSION & CONCLUSION

We find that several syntactic agreement phenomena can be localised to a small number of MLP neurons. This localisation has similar characteristics to the localisation of factual information, suggesting that recent transformer-based language models' impressive abilities with respect to various linguistic phenomena and the recall of facts from their

Table 1: Results obtained under our new criteria suggest model editing methods are not robust.

| Model | Data | Reliability | Measure | |
|---|---|---|---|---|
| GPT-2 XL | P101 | 99.82% | Synonym | 52.35% |
| | P1376 | 96.37% | Symmetry | 23.71% |
| | P36 | 99.79% | Symmetry | 25.17% |
| LLaMA-2 | P101 | 100% | Synonym | 58.36% |
| | P1376 | 100% | Symmetry | 33.40% |
| | P36 | 100% | Symmetry | 33.64% |

training corpora may follow the same underlying mechanism. The localisation of the two types of information also faces the same challenges, however, which militate against the soundness of the KN thesis. Specifically, the effect of editing the identified neurons is not strong enough to overturn the final prediction, and the scope of the phenomena appears to be limited to shallow cues such as token co-occurrence statistics.

Returning to Geva et al.'s (2021) original findings, the MLP neurons store patterns that are interpretable through a linguistic lens, but they do not store knowledge, either linguistic or factual. Meng et al.'s (2022) causal tracing results, although still an oversimplification, suggest that there are different phases in different layers in the entire process of token expression. But their ROME model-editing method did not avail itself of this important finding. The method is still MLP-based. To achieve a better understanding of this expression process and achieve real model editing, we must examine the entire decision-making circuit (Wang et al., 2022; Wu et al., 2023; Conmy et al., 2023; Murty et al., 2023). Manipulating only the MLP weights is not enough. The circuit mode of interpretation is still at a very early state of development, however. Current circuit identification methods are *ad hoc*, furthermore, and have only been applied to a small set of tasks. In future work, we will try to formalize the circuit interpretation framework and apply it to more tasks and phenomena.

Our reassessment of causal traces agrees with Hase et al.'s (2023), but we take exception to their claim that "better mechanistic understanding . . . may not always translate to insights about how to best change their behavior." It is well-established that we can interpret the computational mechanism of LMs through the lens of formal linguistics (Clark et al., 2019). Both of our findings reveal limitations in current LM interpretation work and suggest that an even more comprehensive, but still mechanistic interpretation of transformers will lead to insights for better control of model behaviour when not limited to the MLP modules, and when patterns of token expression are dealt with unencumbered by misbegotten metaphors about knowledge and human reasoning.

**Contributions** Our work provides a thorough examination of the KN thesis and finds that the thesis is, at best, an oversimplification. We (1) extend KN-based analysis to well-defined syntactic tasks, (2) propose two new criteria for evaluating the effectiveness of model editing, and (3) introduce a generalised $n$-sample similarity measure of the level of localisation.

ACKNOWLEDGMENTS

We thank Yu Lei (University of Toronto) for a great deal of insightful discussion. We also want to thank the anonymous reviewers for providing informative comments and suggestions.

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

Table 2: BLiMP phenomena and paradigms.

| Phenomenon | Paradigms | Example |
|---|---|---|
| Anaphor Agreement | anaphor_gender_agreement
anaphor_number_agreement | Katherine can't help herself/*himself.
Many teenagers were helping themselves/*herself. |
| Determiner-Noun Agreement | determiner_noun_agreement_1
determiner_noun_agreement_2
determiner_noun_agreement_irregular_1
determiner_noun_agreement_irregular_2
determiner_noun_agreement_with_adj_1
determiner_noun_agreement_with_adj_2
determiner_noun_agreement_with_adj_irregular_1
determiner_noun_agreement_with_adj_irregular_2 | Craig explored that grocery store/*grocery stores.
Carl cures those/*that horses.
Phillip was lifting this mouse/*this mice.
Those ladies walk through those/*that oases.
Tracy praises those lucky guys/*guy.
Some actors buy these/*this gray books.
This person shouldn't criticize this upset child/*children.
That adult has brought that/*those purple octopus. |
| Subject-Verb Agreement | distractor_agreement_relational_noun
distractor_agreement_relative_clause
irregular_plural_subject_verb_agreement_1
irregular_plural_subject_verb_agreement_2
regular_plural_subject_verb_agreement_1
regular_plural_subject_verb_agreement_2 | A sketch of lights doesn't/*don't appear.
Boys that aren't disturbing Natalie suffer/*suffers.
This goose isn't/*weren't bothering Edward.
The woman/*women cleans every public park.
Jeffrey hasn't/*haven't criticized Donald.
The dress/*dresses crumples. |

# A  EXPERIMENTAL SETUP

## A.1  LANGUAGE MODELS

We experiment on BERT (Devlin et al., 2019), GPT-2 (Radford et al., 2019), and LLaMA-2 (Touvron et al., 2023). We use the `bert-base-cased` version of BERT, the base version GPT-2 and 7B parameter version of LLaMA-2 in Section 3. We added GPT-2 XL in Section 4 as it is also used by (Meng et al., 2022). We use the huggingface package (Wolf et al., 2020) for the implementation.

We choose BERT and GPT-2 as they are the most widely studied and applied language models. We choose LLaMA-2 as a representative of the recent large language models. All three models are transformer-based. BERT is a masked language model (MLM), and GPT-2 and LLaMA are decoder-only language models. MLM is a type of bidirectional language model. It can process context in both the forward and backward direction and the order between source and target is not required. However, decoder-only language models only process context from left to right and this means that the target must locate at the end of the prompt. Therefore, some linguistic patterns are not suitable for decoder-only LMs. We discarded these patterns for GPT-2 and LLaMA.

For the sake of consistency with Meng et al. (2022), we use GPT-XL for causal tracing and the evaluation of ROME. We also evaluate LLaMA-2-7B on ROME as a representative of recent LLMs. Yao et al. (2023) provided a recipe of applying ROME on LLaMA-2-7b,[6] we follow their instructions and hyperparameters to conduct our evaluation.

## A.2  CORPORA

**BLiMP**  We use the linguistic phenomena collected in BLiMP (Warstadt et al., 2020) for our analysis. The BLiMP corpus contains minimal pairs for 12 grammar phenomena. Some of the phenomena are not suitable for our experiments and are therefore discarded. The remaining phenomena and paradigms are shown in Table 2.

**ParaRel**  The corpus PARAREL (Elazar et al., 2021) contains facts formulated as a fill-in-the-blank cloze task and it is curated by experts. It contains 38 relation types and Table 3 provides an overview of an overview of the PARAREL corpus, and 27,738 relational facts in total. We obtain prompts from PARAREL following Dai et al.'s (2022) instructions. For each PARAREL relations, Dai et al. (2022) created multiple prompt templates. On average, they created 8.63 different prompt templates for each of the relations. In total, the PARAREL corpus contains 253,448 prompts.

---

[6]https://github.com/zjunlp/EasyEdit/blob/main/hparams/ROME/llama-7b.yaml

Table 3: An overview of PARAREL relations. There are two bijective relations: P1376 (capital of) and P36 (capital). Both relations are underlined.

| ID | Relation | Relation type |
|---|---|---|
| P1001 | applies to jurisdiction | N-M |
| P101 | field of work | N-M |
| P103 | native language | N-1 |
| P106 | occupation | N-M |
| P108 | employer | N-M |
| P127 | owned by | N-1 |
| P1303 | instrument | N-M |
| P131 | located in the administrative territorial entity | N-1 |
| P136 | genre | N-1 |
| P1376 | capital of | 1-1 |
| P138 | named after | N-1 |
| P140 | religion | N-1 |
| P1412 | languages spoken, written or signed | N-M |
| P159 | headquarters location | N-1 |
| P17 | country | N-1 |
| P176 | manufacturer | N-1 |
| P178 | developer | N-M |
| P19 | place of birth | N-1 |
| P190 | twinned administrative body | N-M |
| P20 | place of death | N-1 |
| P264 | record label | N-1 |
| P27 | country of citizenship | N-M |
| P276 | location | N-1 |
| P279 | subclass of | N-1 |
| P30 | continent | N-1 |
| P36 | capital | 1-1 |
| P361 | part of | N-1 |
| P364 | original language of film or TV show | N-1 |
| P37 | official language | N-1 |
| P39 | position held | N-M |
| P407 | language of work or name | N-1 |
| P413 | position played on team / speciality | N-1 |
| P449 | original network | N-1 |
| P463 | member of | N-M |
| P47 | shares border with | N-M |
| P495 | country of origin | N-1 |
| P530 | diplomatic relation | N-M |
| P740 | location of formation | N-1 |
| P937 | work location | N-M |

There are two bijective (1-1) relations: P1376 (capital of) and P36 (capital). We use those two relations for our bijection reversal relation evaluation.

**CounterFact and ZsRE**  Meng et al. (2022) processed PARAREL relations differently from Dai et al. (2022). In particular, they did not create the new prompt templates for each relational facts.

The Zero-Shot Relation Extraction (zsRE) corpus used by Mitchell et al. (2022); De Cao et al. (2021) is another popular corpus used to evaluate model editing methods. The evaluation slice contains 10,000 records.

Both corpora are used by Meng et al. (2022) and Yao et al. (2023). We did not use these two corpus in our experiments, however, we are citing results conducted by Meng et al. (2022) and Yao et al. (2023) to avoid duplication. Our analysis of factual information can be easily generalised to these two corpora and we plan to expand our analysis to more corpora for future work.

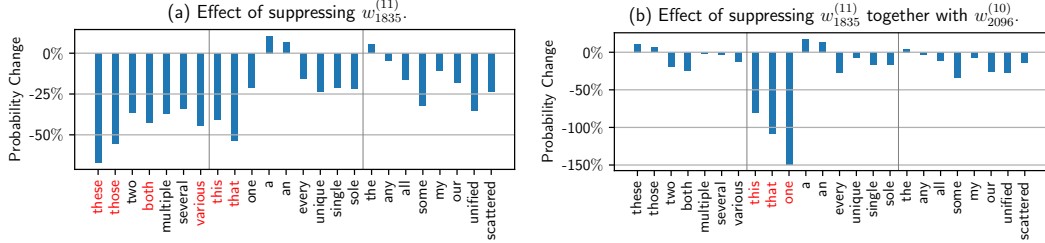

Figure 9: Despite $w_{1835}^{11}$'s high occurrence in Det-N pairs with *this* and *that*, because its 3% strong appearance in Det-N pairs with *these*, it is not a singular grammatical number KN.

# B  DETERMINER GRAMMATICAL NUMBER KN SEARCH

## B.1  PRENOMINAL MODIFIERS

We study the following prenominal modifiers:

- Determiners:
    - The demonstrative determiners used by BLiMP: *this, that, these, those*;
    - Plural determiners: *two, both, multiple, several, various*;
    - Singular determiners: *one, a, an, every*;
    - Determiners that do not express number agreement: *the, some, any, all, my, our*;
- Adjectives: *single, unique, sole, scattered, unified*.

The six modifiers that do not express grammatical number agreement: *any, all, my, our scattered* and *unified* are underlined. However, these modifiers are used more often with numbered nouns because of their meanings.

## B.2  THE KN SEARCH

Table 4 shows the full list of neurons identified for the paradigm. We identified two knowledge neurons. The plural neuron $w_{1094}^{9}$ is highlighted in **blue** and the singular neuron $w_{2096}^{10}$ is highlighted in **red**.

The neuron $w_{1835}^{11}$ is an interesting case. It appears as a knowledge neuron in 92% and 90% of the determiner *this* and *that*, and only 3% of the determiner *these*. However, these 3% of the neurons are very strong. Including $w_{1835}^{11}$ as a singular neuron, or using the neuron on its own does not should good localisation of grammatical number (Figure 9). Therefore, for our KN search, we excluded these neurons.

Table 4: Neurons identified using BLiMP's determiner_noun_agreement_2 paradigm.

| Neuron | this | that | these | those |
|---|---|---|---|---|
| $w_{1094}^9$ | 0.00 | 0.00 | 1.00 | 1.00 |
| $w_{1835}^{11}$ | 0.92 | 0.90 | 0.03 | 0.00 |
| $w_{2096}^{10}$ | 0.88 | 0.81 | 0.00 | 0.00 |
| $w_{2686}^{11}$ | 0.00 | 0.85 | 0.00 | 0.04 |
| $w_{2339}^9$ | 0.27 | 0.03 | 0.36 | 0.00 |
| $w_{2539}^{10}$ | 0.00 | 0.00 | 0.14 | 0.00 |
| $w_{2999}^{10}$ | 0.04 | 0.00 | 0.12 | 0.00 |
| $w_{10}^{11}$ | 0.01 | 0.00 | 0.07 | 0.03 |
| $w_{633}^{10}$ | 0.00 | 0.00 | 0.07 | 0.00 |
| $w_{651}^{10}$ | 0.00 | 0.00 | 0.07 | 0.09 |
| $w_{2231}^{11}$ | 0.07 | 0.00 | 0.00 | 0.00 |
| $w_{2029}^{10}$ | 0.00 | 0.00 | 0.06 | 0.06 |
| $w_{866}^{11}$ | 0.03 | 0.03 | 0.06 | 0.03 |
| $w_{141}^{10}$ | 0.00 | 0.00 | 0.04 | 0.01 |
| $w_{1405}^{11}$ | 0.04 | 0.01 | 0.00 | 0.00 |
| $w_{3034}^7$ | 0.00 | 0.00 | 0.03 | 0.00 |
| $w_{900}^{11}$ | 0.03 | 0.01 | 0.03 | 0.00 |
| $w_{2606}^{10}$ | 0.00 | 0.00 | 0.03 | 0.00 |
| $w_{1723}^{11}$ | 0.03 | 0.00 | 0.00 | 0.66 |
| $w_{35}^{10}$ | 0.03 | 0.01 | 0.00 | 0.01 |
| $w_{1404}^8$ | 0.03 | 0.00 | 0.00 | 0.00 |
| $w_{646}^{11}$ | 0.00 | 0.01 | 0.00 | 0.00 |
| $w_{1626}^{10}$ | 0.00 | 0.01 | 0.00 | 0.00 |
| $w_{412}^9$ | 0.00 | 0.01 | 0.00 | 0.00 |
| $w_{2123}^6$ | 0.00 | 0.00 | 0.01 | 0.00 |
| $w_{2766}^9$ | 0.00 | 0.00 | 0.01 | 0.01 |
| $w_{1845}^{10}$ | 0.00 | 0.00 | 0.01 | 0.00 |
| $w_{1248}^{11}$ | 0.00 | 0.00 | 0.01 | 0.00 |
| $w_{444}^{11}$ | 0.01 | 0.00 | 0.00 | 0.00 |
| $w_{1248}^7$ | 0.01 | 0.00 | 0.00 | 0.00 |
| $w_{2480}^{11}$ | 0.01 | 0.00 | 0.00 | 0.01 |
| $w_{1824}^{11}$ | 0.01 | 0.00 | 0.00 | 0.00 |
| $w_{2754}^8$ | 0.00 | 0.00 | 0.00 | 0.06 |
| $w_{606}^{10}$ | 0.00 | 0.00 | 0.00 | 0.01 |
| $w_{602}^6$ | 0.00 | 0.00 | 0.00 | 0.01 |
| $w_{175}^{10}$ | 0.00 | 0.00 | 0.00 | 0.10 |
| $w_{1568}^{11}$ | 0.00 | 0.00 | 0.00 | 0.01 |

## C  KNs FOR LINGUISTIC PHENOMENA

In this section, we are going to present the KN search for other linguistic phenomena and models. Table 5, 6 and 7 lists all the neurons we identified for each of the BLiMP paradigms and language models. Some of the paradigms such as determiner_noun_agreement_2 are not applicable for decoder only language models because the target precedes the source. We do not evaluate these paradigms for the two decoder-only language models.

Similar to the search for determiner_noun_agreement_2 KNs, we first identify neurons with common appearance in the prompts with a certain grammatical property. Then, we manually test if suppressing these neurons can lead to significant model behaviour change. The number of neurons we identified is different across model and paradigms. The exact pattern behind this difference may need more investigation. We leave it for future work.

Please consult our code release repository[7] for any updates, detailed analysis and documentation of these KN search results.

---

[7]https://github.com/frankniujc/kn_thesis

Table 5: BLiMP phenomena and paradigms.

| Phenomenon | Paradigms | Property | Model | KNs |
|---|---|---|---|---|
| Anaphor Agreement | anaphor_gender _agreement | m | BERT | $w_{942}^7, w_{2881}^8, w_{1845}^{10}$ |
| | | | GPT-2 | $w_{2985}^9, w_{17}^{11}, w_{1611}^{11}, w_{2044}^{11}, w_{2910}^{11}$ |
| | | | LLaMA-2 | $w_{6454}^0, w_{5279}^{30}, w_{10638}^{31}$ |
| | | f | BERT | $w_{942}^7, w_{1712}^9$ |
| | | | GPT-2 | $w_{1344}^0, w_{1403}^0, w_{1253}^8, w_{1891}^8, w_{2093}^{10}$ |
| | | | LLaMA-2 | $w_{10442}^{29}, w_{3882}^{30}, w_{6935}^{30}, w_{2606}^{31}, w_{7984}^{31}$ |
| | anaphor_number _agreement | sg | BERT | $w_{1712}^9$ |
| | | | GPT-2 | $w_{1891}^8, w_{1690}^{10}$ |
| | | | LLaMA-2 | $w_{5279}^{30}, w_{7839}^{31}, w_{9148}^{31}$ |
| | | pl | BERT | $w_{2070}^{11}$ |
| | | | GPT-2 | $w_{3060}^{10}, w_{598}^{11}$ |
| | | | LLaMA-2 | $w_{1116}^{31}, w_{6124}^{31}, w_{7742}^{31}, w_{8169}^{31}$ |
| Determiner-Noun Agreement | determiner_noun _agreement_1 | sg | BERT | $w_{452}^7, w_{2631}^7, w_{1222}^8, w_{2660}^8, w_{620}^9, w_{1283}^9, w_{136}^{10}, w_{598}^{10}, w_{1038}^{10},$ $w_{1143}^{10}, w_{1279}^{10}, w_{1418}^{10}, w_{2162}^{10}, w_{384}^{11}, w_{526}^{11}, w_{558}^{11}, w_{762}^{11}, w_{870}^{11},$ $w_{991}^{11}, w_{999}^{11}, w_{1143}^{11}, w_{1267}^{11}, w_{1350}^{11}, w_{1435}^{11}, w_{1496}^{11}, w_{1521}^{11}, w_{1565}^{11},$ $w_{2204}^{11}, w_{2221}^{11}, w_{2592}^{11}, w_{2632}^{11}, w_{2772}^{11}, w_{2847}^{11}, w_{2994}^{11}$ |
| | | | GPT-2 | $w_{11}^0, w_{37}^0, w_{132}^0, w_{1294}^0, w_{1311}^0, w_{1414}^0, w_{1529}^0, w_{1797}^0, w_{1950}^0,$ $w_{2577}^0, w_{2733}^0, w_{2367}^7, w_{571}^{10}, w_{1998}^{11}$ |
| | | | LLaMA-2 | $w_{5279}^{30}, w_{8177}^{30}, w_{1876}^{31}, w_{5591}^{31}, w_{5876}^{31}, w_{8061}^{31}, w_{8236}^{31}$ |
| | | pl | BERT | $w_{52}^8, w_{218}^9, w_{698}^9, w_{1343}^9, w_{1812}^9, w_{2158}^9, w_6^{10}, w_{845}^{10}, w_{883}^{10}, w_{975}^{10},$ $w_{1178}^{10}, w_{1959}^{10}, w_9^{11}, w_{199}^{11}, w_{310}^{11}, w_{532}^{11}, w_{631}^{11}, w_{1009}^{11}, w_{1040}^{11},$ $w_{1396}^{11}, w_{1548}^{11}, w_{1767}^{11}, w_{1965}^{11}, w_{1985}^{11}, w_{2646}^{11}, w_{2978}^{11}, w_{2995}^{11}$ |
| | | | GPT-2 | $w_{646}^0, w_{1013}^0, w_{1159}^0, w_{1227}^0, w_{1382}^0, w_{1469}^0, w_{1612}^0, w_{2428}^0, w_{2702}^0,$ $w_{3055}^0, w_{1871}^7, w_{476}^{10}, w_{1693}^{10}, w_{3038}^{10}, w_{472}^{11}, w_{2387}^{11}$ |
| | | | LLaMA-2 | $w_{7003}^2, w_{4435}^5, w_{6139}^{15}, w_{376}^{30}, w_{2262}^{30}, w_{3619}^{30}, w_{4228}^{30}, w_{4257}^{30}, w_{7935}^{30},$ $w_{9673}^{30}$ |
| | determiner_noun _agreement_2 | sg | BERT | $w_{2096}^{10}$ |
| | | pl | BERT | $w_{1094}^9$ |
| | determiner_noun _agreement_irregular_1 | sg | BERT | $w_{655}^7, w_{656}^8, w_{1222}^8, w_{561}^9, w_{1146}^9, w_{2393}^9, w_{136}^{10}, w_{598}^{10}, w_{1143}^{10},$ $w_{1418}^{10}, w_{1955}^{10}, w_{237}^{11}, w_{281}^{11}, w_{326}^{11}, w_{558}^{11}, w_{1309}^{11}, w_{1350}^{11}, w_{1521}^{11},$ $w_{2257}^{11}, w_{2388}^{11}, w_{2592}^{11}, w_{2870}^{11}, w_{2994}^{11}, w_{3020}^{11}$ |
| | | | GPT-2 | $w_{11}^0, w_{2845}^0, w_{628}^6, w_{2216}^6, w_{569}^9, w_{2603}^9, w_{383}^{10}, w_{534}^{10}, w_{571}^{10}$ |
| | | | LLaMA-2 | $w_{168}^{29}, w_{3619}^{30}, w_{3937}^{30}, w_{8298}^{30}, w_{5591}^{31}, w_{8169}^{31}, w_{10905}^{31}$ |
| | | pl | BERT | $w_{2594}^8, w_{698}^9, w_{1106}^9, w_{2158}^9, w_{664}^{10}, w_{845}^{10}, w_{1178}^{10}, w_{1547}^{10}, w_{2810}^{10},$ $w_{26}^{11}, w_{45}^{11}, w_{310}^{11}, w_{532}^{11}, w_{631}^{11}, w_{1239}^{11}, w_{1873}^{11}, w_{1934}^{11}, w_{1965}^{11}, w_{2070}^{11},$ $w_{2320}^{11}, w_{2944}^{11}, w_{2978}^{11}, w_{2995}^{11}$ |
| | | | GPT-2 | $w_{185}^0, w_{280}^0, w_{1871}^7, w_{55}^8, w_{792}^9, w_{1693}^{10}, w_{3038}^{10}, w_{472}^{11}, w_{2387}^{11}$ |
| | | | LLaMA-2 | $w_{1443}^{30}, w_{1686}^{30}, w_{3397}^{30}, w_{7455}^{30}, w_{8343}^{30}, w_{8878}^{30}, w_{9673}^{30}, w_{6587}^{31}$ |
| | determiner_noun _agreement_irregular_2 | sg | BERT | $w_{2096}^{10}$ |
| | | pl | BERT | $w_{1094}^9$ |

Table 6: BLiMP phenomena and paradigms (continued).

| Phenomenon | Paradigms | Property | Model | KNs |
|---|---|---|---|---|
| Determiner-Noun Agreement (continued) | determiner_noun_agreement_with_adj_1 | sg | BERT | $w_{64}^{7}, w_{984}^{7}, w_{391}^{9}, w_{1283}^{9}, w_{2381}^{9}, w_{2951}^{9}, w_{995}^{10}, w_{1235}^{10}, w_{1269}^{10}, w_{1279}^{10}, w_{1382}^{10}, w_{1737}^{10}, w_{1955}^{10}, w_{2024}^{10}, w_{2935}^{10}, w_{74}^{11}, w_{558}^{11}, w_{991}^{11}, w_{1350}^{11}, w_{1521}^{11}, w_{2173}^{11}, w_{2647}^{11}$ |
| | | | GPT-2 | $w_{1344}^{0}, w_{1888}^{5}, w_{1871}^{7}, w_{1253}^{8}, w_{330}^{9}, w_{80}^{10}, w_{379}^{10}, w_{383}^{10}, w_{2178}^{10}, w_{2738}^{10}, w_{740}^{11}, w_{1593}^{11}, w_{2044}^{11}$ |
| | | | LLaMA-2 | $w_{5279}^{30}, w_{8177}^{30}, w_{1876}^{31}, w_{5591}^{31}, w_{5876}^{31}, w_{8061}^{31}, w_{8236}^{31}$ |
| | | pl | BERT | $w_{3052}^{8}, w_{698}^{9}, w_{1343}^{9}, w_{1812}^{9}, w_{2158}^{9}, w_{2327}^{9}, w_{6}^{10}, w_{664}^{10}, w_{845}^{10}, w_{1178}^{10}, w_{1624}^{10}, w_{1888}^{10}, w_{1959}^{10}, w_{122}^{11}, w_{291}^{11}, w_{310}^{11}, w_{532}^{11}, w_{1009}^{11}, w_{2042}^{11}, w_{2070}^{11}, w_{2106}^{11}, w_{2978}^{11}, w_{2995}^{11}$ |
| | | | GPT-2 | $w_{289}^{3}, w_{1993}^{8}, w_{840}^{9}, w_{54}^{10}, w_{476}^{10}, w_{1693}^{10}, w_{3038}^{10}, w_{713}^{11}, w_{992}^{11}, w_{2387}^{11}, w_{2408}^{11}, w_{2605}^{11}$ |
| | | | LLaMA-2 | $w_{7003}^{2}, w_{4435}^{5}, w_{6139}^{15}, w_{376}^{30}, w_{2262}^{30}, w_{3619}^{30}, w_{4228}^{30}, w_{4257}^{30}, w_{7935}^{30}, w_{9673}^{30}$ |
| | determiner_noun_agreement_with_adj_2 | sg | BERT | $w_{2096}^{10}$ |
| | | pl | BERT | $w_{1094}^{9}$ |
| | determiner_noun_agreement_with_adj_irregular_1 | sg | BERT | $w_{1985}^{8}, w_{391}^{9}, w_{1450}^{9}, w_{2407}^{9}, w_{136}^{10}, w_{247}^{10}, w_{598}^{10}, w_{694}^{10}, w_{1143}^{10}, w_{1955}^{10}, w_{2024}^{10}, w_{155}^{11}, w_{281}^{11}, w_{558}^{11}, w_{1084}^{11}, w_{1224}^{11}, w_{1315}^{11}, w_{1350}^{11}, w_{1521}^{11}, w_{1938}^{11}, w_{3070}^{11}$ |
| | | | GPT-2 | $w_{1344}^{0}, w_{1888}^{5}, w_{1871}^{7}, w_{1253}^{8}, w_{330}^{9}, w_{80}^{10}, w_{379}^{10}, w_{383}^{10}, w_{2178}^{10}, w_{2738}^{10}, w_{740}^{11}, w_{1593}^{11}, w_{2044}^{11}$ |
| | | | LLaMA-2 | $w_{7003}^{2}, w_{3937}^{30}, w_{6935}^{30}, w_{8298}^{30}, w_{9131}^{30}, w_{7988}^{31}, w_{8236}^{31}$ |
| | | pl | BERT | $w_{426}^{8}, w_{698}^{9}, w_{2158}^{9}, w_{845}^{10}, w_{933}^{10}, w_{1178}^{10}, w_{1911}^{11}, w_{165}^{11}, w_{211}^{11}, w_{307}^{11}, w_{310}^{11}, w_{532}^{11}, w_{662}^{11}, w_{1107}^{11}, w_{1177}^{11}, w_{1548}^{11}, w_{1934}^{11}, w_{1965}^{11}, w_{2070}^{11}, w_{2106}^{11}, w_{2621}^{11}, w_{2704}^{11}, w_{2944}^{11}, w_{2995}^{11}, w_{3037}^{11}$ |
| | | | GPT-2 | $w_{2702}^{0}, w_{1993}^{8}, w_{54}^{10}, w_{900}^{10}, w_{1693}^{10}, w_{3038}^{10}, w_{472}^{11}, w_{2387}^{11}, w_{2408}^{11}, w_{2605}^{11}$ |
| | | | LLaMA-2 | $w_{5180}^{15}, w_{8878}^{30}, w_{10417}^{30}, w_{10552}^{30}, w_{10588}^{30}, w_{6587}^{31}, w_{11000}^{31}$ |
| | determiner_noun_agreement_with_adj_irregular_2 | sg | BERT | $w_{2096}^{10}$ |
| | | pl | BERT | $w_{1094}^{9}$ |
| Subject-Verb Agreement | regular_plural_subject_verb_agreement_1 | sg | BERT | $w_{455}^{10}, w_{153}^{11}, w_{1541}^{11}$ |
| | | | GPT-2 | $w_{379}^{10}, w_{729}^{10}, w_{2839}^{10}, w_{2173}^{11}, w_{2187}^{11}$ |
| | | | LLaMA-2 | $w_{566}^{31}$ |
| | | pl | BERT | $w_{1073}^{11}, w_{1307}^{11}$ |
| | | | GPT-2 | $w_{101}^{0}, w_{2674}^{6}, w_{1318}^{10}, w_{1347}^{10}, w_{2409}^{10}, w_{17}^{11}, w_{627}^{11}, w_{896}^{11}, w_{1043}^{11}, w_{3051}^{11}$ |
| | | | LLaMA-2 | $w_{4054}^{17}, w_{5279}^{30}, w_{556}^{31}$ |
| | regular_plural_subject_verb_agreement_2 | sg | BERT | $w_{1350}^{11}$ |
| | | pl | BERT | $w_{1178}^{10}, w_{2995}^{11}$ |
| | irregular_plural_subject_verb_agreement_1 | sg | BERT | $w_{455}^{10}, w_{153}^{11}, w_{1541}^{11}$ |
| | | | GPT-2 | $w_{379}^{10}, w_{729}^{10}, w_{2839}^{10}, w_{2173}^{11}, w_{2187}^{11}$ |
| | | | LLaMA-2 | $w_{566}^{31}$ |
| | | pl | BERT | $w_{1073}^{11}, w_{1307}^{11}$ |
| | | | GPT-2 | $w_{101}^{0}, w_{2674}^{6}, w_{1318}^{10}, w_{1347}^{10}, w_{2409}^{10}, w_{17}^{11}, w_{627}^{11}, w_{896}^{11}, w_{1043}^{11}, w_{3051}^{11}$ |
| | | | LLaMA-2 | $w_{5279}^{30}, w_{212}^{31}, w_{2606}^{31}, w_{7839}^{31}$ |
| | irregular_plural_subject_verb_agreement_2 | sg | BERT | $w_{1350}^{11}$ |
| | | pl | BERT | $w_{1178}^{10}, w_{2995}^{11}$ |

Table 7: BLiMP phenomena and paradigms (continued).

| Phenomenon | Paradigms | Property | Model | KNs |
|---|---|---|---|---|
| Subject-Verb Agreement (continued) | distractor_agreement _relational_noun | sg | BERT | $w^{10}_{455}, w^{11}_{153}, w^{11}_{1541}$ |
| | | | GPT-2 | $w^{10}_{379}, w^{10}_{729}, w^{10}_{2839}, w^{11}_{2173}, w^{11}_{2187}$ |
| | | | LLaMA-2 | $w^{31}_{566}$ |
| | | pl | BERT | $w^{9}_{2253}, w^{11}_{1307}$ |
| | | | GPT-2 | $w^{0}_{101}, w^{6}_{2674}, w^{10}_{1318}, w^{10}_{1347}, w^{10}_{2409}, w^{11}_{17}, w^{11}_{627}, w^{11}_{896}, w^{11}_{1043}, w^{11}_{3051}$ |
| | | | LLaMA-2 | $w^{30}_{5279}, w^{31}_{3336}, w^{31}_{3658}, w^{31}_{7342}$ |
| | distractor_agreement _relative_clause | sg | BERT | $w^{11}_{798}, w^{11}_{1541}$ |
| | | | GPT-2 | $w^{10}_{379}, w^{10}_{729}, w^{10}_{2839}, w^{11}_{2173}, w^{11}_{2187}$ |
| | | | LLaMA-2 | $w^{31}_{566}$ |
| | | pl | BERT | $w^{9}_{2253}$ |
| | | | GPT-2 | $w^{0}_{101}, w^{6}_{2674}, w^{10}_{1318}, w^{10}_{1347}, w^{10}_{2409}, w^{11}_{17}, w^{11}_{627}, w^{11}_{896}, w^{11}_{1043}, w^{11}_{3051}$ |
| | | | LLaMA-2 | $w^{30}_{5279}, w^{31}_{212}, w^{31}_{2606}, w^{31}_{7839}$ |

## C.1 EFFECTS OF SUPPRESSING THE KNS

Figure 10 shows the probability change after erasing the identified singular or plural neuron. All the results are similar to the base case paradigm presented in the main section of the paper. We can observe similar levels of probability change as determiner_noun_agreement_2 with or without adding distractors (adjectives and irregular verbs).

Figure 11 shows the probability change of subject-verb agreement paradigms on GPT-2 and LLaMA-2. For GPT-2, we can see that the suppression of the singular neurons causes significant probability decrease for singular verbs and essentially no effect to plural verbs. The effect of intervention is more pronounced for the plural neurons. For LLaMA-2, however, as the model becomes larger, the relative importance of each neuron becomes smaller. Therefore, we observe that the effect of KN editing is less pronounced.

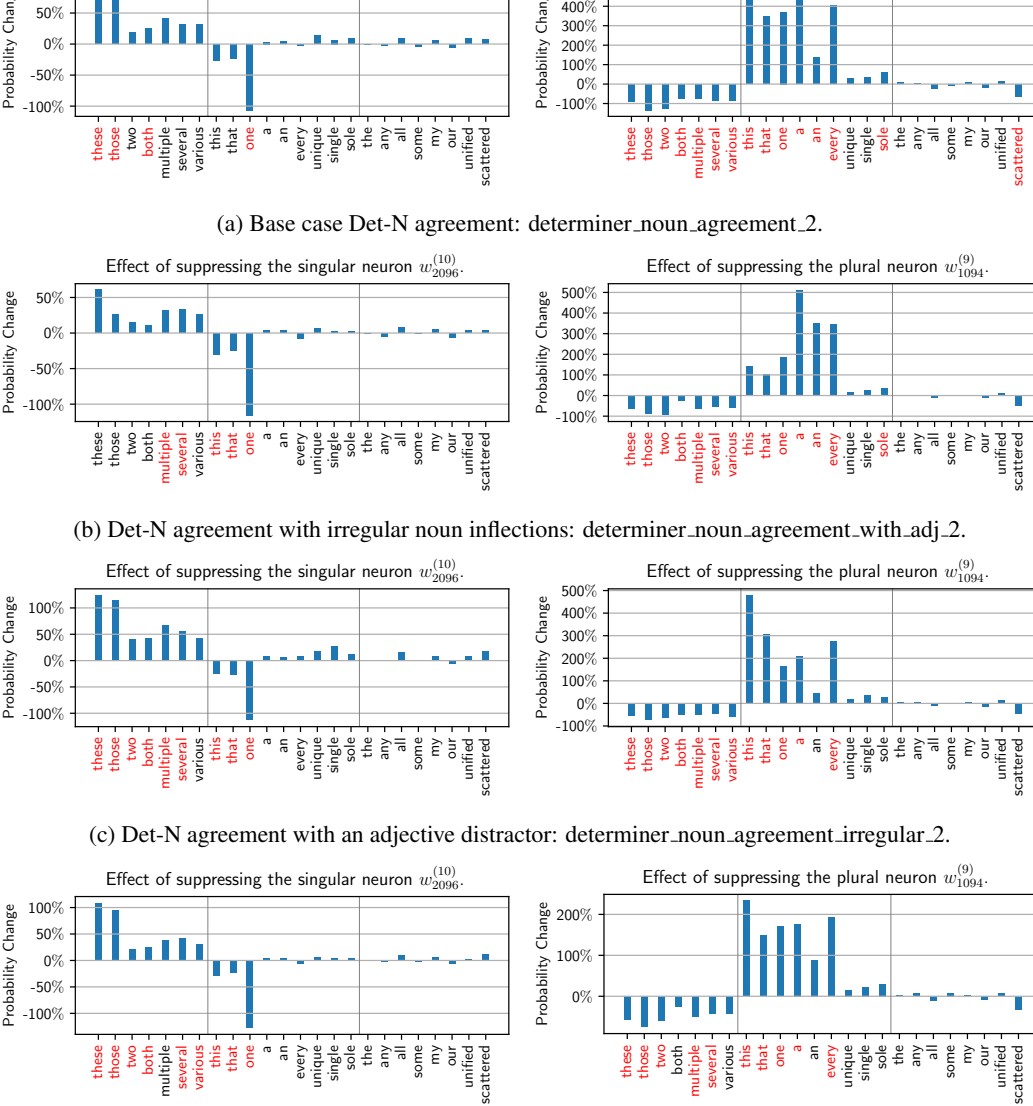

(a) Base case Det-N agreement: determiner_noun_agreement_2.

(b) Det-N agreement with irregular noun inflections: determiner_noun_agreement_with_adj_2.

(c) Det-N agreement with an adjective distractor: determiner_noun_agreement_irregular_2.

(d) Adjective distractor and irregular nouns: determiner_noun_agreement_with_adj_irregular_2

Figure 10: Effect of suppressing the KNs of type 2 determiner-noun agreement on BERT. We can observe similar levels of probability change as determiner_noun_agreement_2 with or without adding distractors (adjectives and irregular verbs).

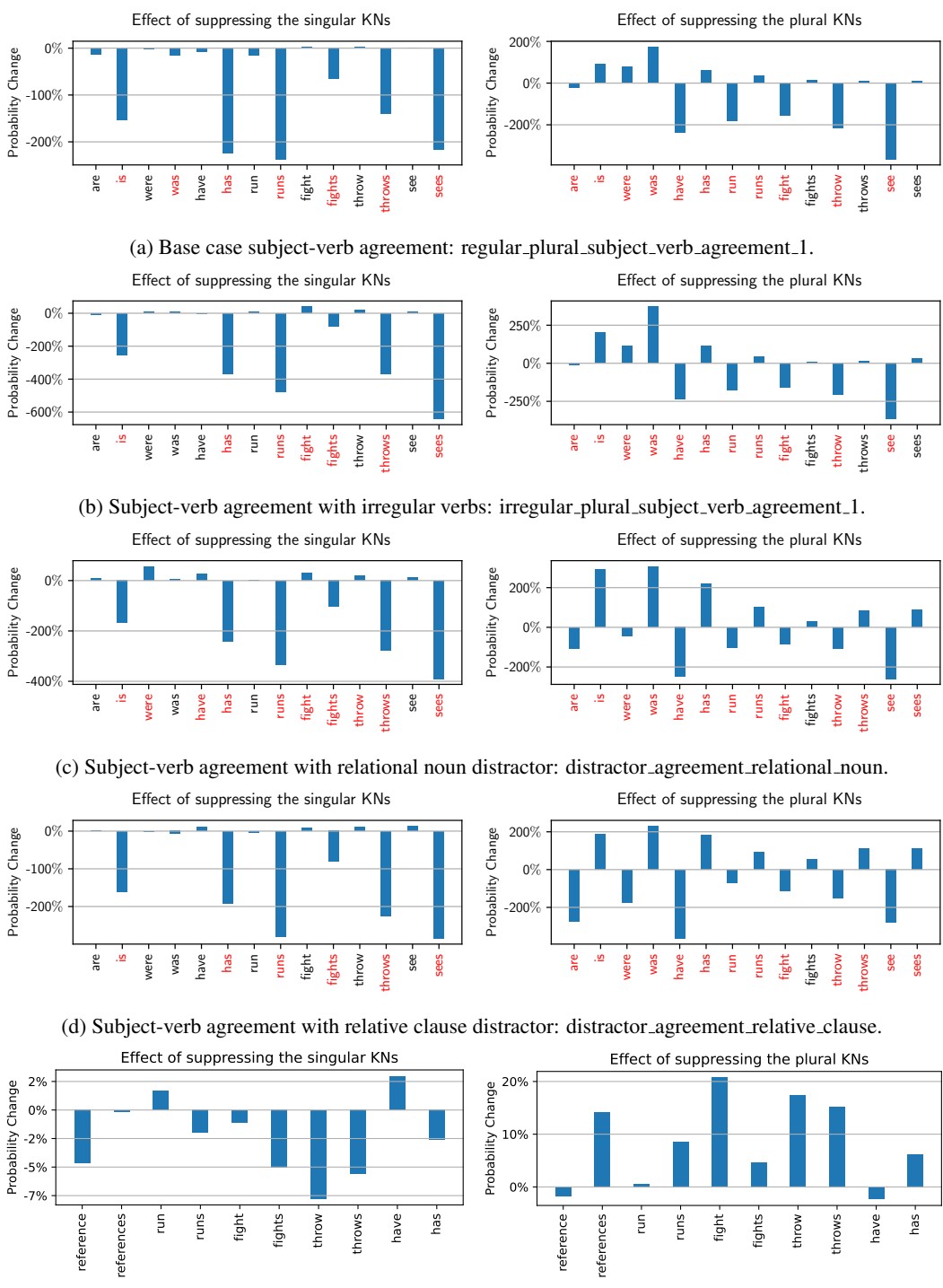

(a) Base case subject-verb agreement: regular_plural_subject_verb_agreement_1.

(b) Subject-verb agreement with irregular verbs: irregular_plural_subject_verb_agreement_1.

(c) Subject-verb agreement with relational noun distractor: distractor_agreement_relational_noun.

(d) Subject-verb agreement with relative clause distractor: distractor_agreement_relative_clause.

(e) LLaMA-2 results on regular_plural_subject_verb_agreement_1. The effect of editing is less pronounced on larger LMs, as each neuron has smaller relative importance.

Figure 11: Effect of suppressing the subject-verb agreement KNs on GPT-2.

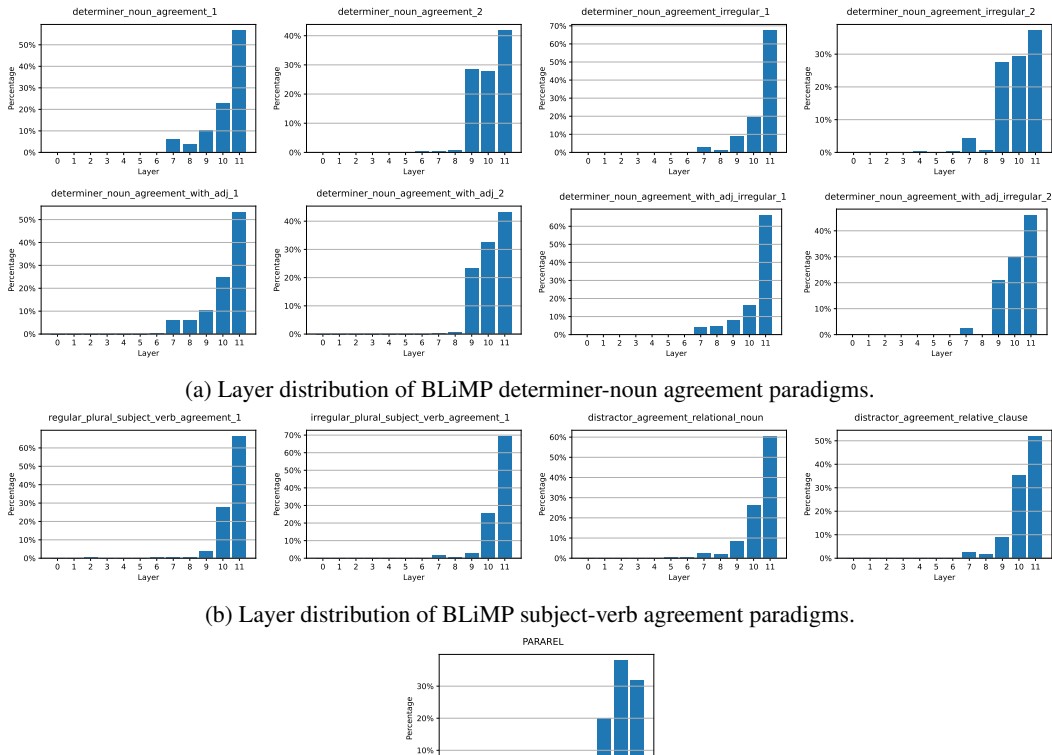

(a) Layer distribution of BLiMP determiner-noun agreement paradigms.

(b) Layer distribution of BLiMP subject-verb agreement paradigms.

(c) Recreation of Dai et al.'s (2022) KN distribution result across ParaRel relations (Figure 3).

Figure 12: Percentage of knowledge neurons identified in different BLiMP paradigms using BERT.

## C.2 LAYER DISTRIBUTION OF IDENTIFIED KNS

In Figure 12, we analysed the layers of the identified KNs on each BLiMP paradigm and compare it with Dai et al.'s (2022) finding on PARAREL. We notice that the vast majority of neurons of all types are distributed in the topmost layers of BERT. This result confirms Dai et al.'s (2022) observation but disproved their position. There is nothing unique to fact-related neurons. This finding agrees with Niu et al.'s (2022) refutation of Jawahar et al. (2019) and Tenney et al. (2019). Syntactic information (formal competence) and semantic information (functional competence) do not occupy different layers of the language model.

## C.3 LEVELS OF LOCALISATION

Table 8 shows the levels of localisation on BERT across all determiner-noun agreement (DNA) BLiMP paradigms. These measures are comparable to PARAREL results shown in Table 9.

Table 8: Levels of localisation of determiner-noun agreement.

| Paradigm | BERT | | | GPT-2 | | | LLaMA-2 | | |
|---|---|---|---|---|---|---|---|---|---|
| | \|KN\| | $\tau$ | $R_1^2$ | \|KN\| | $\tau$ | $R_1^2$ | \|KN\| | $\tau$ | $R_1^2$ |
| determiner_noun_agreement_1 | 3.94 | 0.71 | 0.56 | 0.06 | 0.45 | 0.15 | 3.38 | 0.44 | 0.24 |
| determiner_noun_agreement_2 | 1.86 | 0.62 | 0.56 | - | - | - | - | - | - |
| determiner_noun_agreement_irregular_1 | 5.53 | 0.73 | 0.64 | 1.32 | 0.58 | 0.24 | 4.93 | 0.48 | 0.46 |
| determiner_noun_agreement_irregular_2 | 2.45 | 0.67 | 0.55 | - | - | - | - | - | - |
| determiner_noun_agreement_with_adjective_1 | 8.88 | 0.78 | 0.67 | 1.31 | 0.62 | 0.17 | 3.38 | 0.44 | 0.24 |
| determiner_noun_agreement_with_adjective_2 | 2.26 | 0.67 | 0.57 | - | - | - | - | - | - |
| determiner_noun_agreement_with_adj_irregular_1 | 9.79 | 0.78 | 0.67 | 0.12 | 0.51 | 0.15 | 4.55 | 0.45 | 0.48 |
| determiner_noun_agreement_with_adj_irregular_2 | 2.60 | 0.69 | 0.58 | - | - | - | - | - | - |

Table 9: Levels of localisation of different PARAREL relations.

| Relation | BERT | | | GPT-2 | | | LLaMA-2 | | |
|---|---|---|---|---|---|---|---|---|---|
| | \|KN\| | $\tau$ | $R_1^2$ | \|KN\| | $\tau$ | $R_1^2$ | \|KN\| | $\tau$ | $R_1^2$ |
| P101 | 0.167 | 0.515 | 0.399 | 1.537 | 0.708 | 0.278 | 1.0 | 0.61 | 0.306 |
| P103 | 0.204 | 0.662 | 0.399 | 1.968 | 0.649 | 0.375 | 7.53 | 0.733 | 0.410 |
| P106 | 1.292 | 0.607 | 0.365 | 10.090 | 0.853 | 0.599 | 0.28 | 0.438 | 0.258 |
| P108 | 1.493 | 0.663 | 0.473 | 10.433 | 0.848 | 0.269 | 18.1 | 0.735 | 0.599 |
| P127 | 1.512 | 0.630 | 0.552 | 11.758 | 0.769 | 0.549 | 0.3 | 0.585 | 0.163 |
| P1303 | 10.462 | 0.814 | 0.684 | 12.453 | 0.771 | 0.573 | 0.3 | 0.63 | 0.303 |
| P136 | 14.862 | 0.856 | 0.646 | 14.435 | 0.878 | 0.677 | 2.1 | 0.64 | 0.754 |
| P1376 | 15.640 | 0.842 | 0.628 | 14.892 | 0.794 | 0.624 | 1.4 | 0.592 | 0.435 |
| P138 | 16.992 | 0.958 | 0.874 | 15.365 | 0.794 | 0.621 | 1.4 | 0.62 | 0.605 |
| P140 | 2.008 | 0.689 | 0.263 | 16.543 | 0.848 | 0.707 | 0.7 | 0.64 | 0.290 |
| P1412 | 2.196 | 0.687 | 0.612 | 18.286 | 0.782 | 0.618 | 1.8 | 0.66 | 0.590 |
| P159 | 2.200 | 0.666 | 0.392 | 19.285 | 0.838 | 0.626 | 9.5 | 0.765 | 0.637 |
| P176 | 2.376 | 0.680 | 0.254 | 19.299 | 0.823 | 0.650 | 2.7 | 0.675 | 0.524 |
| P178 | 2.553 | 0.686 | 0.392 | 19.898 | 0.828 | 0.686 | 2.1 | 0.65 | 0.414 |
| P19 | 2.597 | 0.693 | 0.481 | 2.603 | 0.661 | 0.417 | 0.24 | 0.482 | 0.526 |
| P190 | 2.920 | 0.673 | 0.308 | 21.989 | 0.949 | 0.757 | 0.8 | 0.54 | 0.216 |
| P20 | 28.993 | 0.883 | 0.644 | 22.984 | 0.871 | 0.671 | 0.52 | 0.458 | 0.377 |
| P264 | 3.335 | 0.667 | 0.213 | 23.082 | 0.867 | 0.685 | 1.7 | 0.605 | 0.416 |
| P27 | 3.925 | 0.722 | 0.483 | 29.296 | 0.862 | 0.619 | 1.8 | 0.67 | 0.376 |
| P279 | 4.286 | 0.691 | 0.657 | 3.437 | 0.649 | 0.360 | 14.7 | 0.755 | 0.683 |
| P30 | 4.338 | 0.698 | 0.449 | 42.128 | 0.856 | 0.642 | 3.27 | 0.673 | 0.288 |
| P36 | 4.428 | 0.714 | 0.619 | 5.195 | 0.776 | 0.472 | 4 | 0.645 | 0.442 |
| P364 | 4.541 | 0.709 | 0.553 | 5.545 | 0.707 | 0.481 | 3.47 | 0.69 | 0.478 |
| P37 | 4.876 | 0.718 | 0.477 | 5.863 | 0.787 | 0.489 | 0.7 | 0.56 | 0.447 |
| P39 | 5.230 | 0.722 | 0.413 | 6.051 | 0.683 | 0.450 | 0.6 | 0.59 | 0.243 |
| P407 | 5.456 | 0.730 | 0.611 | 6.149 | 0.680 | 0.483 | 0.04 | 0.358 | 0.280 |
| P413 | 5.632 | 0.809 | 0.539 | 6.346 | 0.740 | 0.522 | 0.9 | 0.63 | 0.252 |
| P449 | 6.005 | 0.710 | 0.717 | 7.150 | 0.743 | 0.501 | 0.6 | 0.53 | 0.373 |
| P463 | 6.088 | 0.776 | 0.467 | 7.697 | 0.735 | 0.437 | 1.7 | 0.65 | 0.489 |
| P47 | 6.356 | 0.776 | 0.739 | 8.078 | 0.701 | 0.524 | 11.7 | 0.8 | 0.730 |
| P495 | 6.455 | 0.709 | 0.513 | 8.209 | 0.744 | 0.569 | 1.0 | 0.505 | 0.585 |
| P530 | 7.280 | 0.785 | 0.688 | 8.598 | 0.765 | 0.558 | 1.84 | 0.552 | 0.615 |
| P740 | 7.324 | 0.769 | 0.518 | 8.840 | 0.737 | 0.477 | 2.1 | 0.505 | 0.796 |
| P937 | 8.623 | 0.725 | 0.572 | 9.594 | 0.782 | 0.621 | 2.53 | 0.673 | 0.752 |

# D CAUSAL TRACING

In this section we present an extended analysis of our causal tracing analysis. We use the same hyperparameters and settings as suggested by Meng et al. (2022).

Meng et al.'s (2022) causal tracing experiment is conducted in three steps:

1. In the clean run, they pass a prompt into the model and record all the hidden activation values.

2. Then, they conduct a corrupt run. They perturb the prompt's subject by adding a noise value $\epsilon$ to the input embedding. Because of the obfuscation, the model will like generate an incorrect answer.

3. Finally, in the corrupted-with-restoration run, they let the model run with the corrupted value. But every time for a pair of token and layer, they replace the corrupted output with the original clean state. Then they let the model continue without further intervention. If this restoration can attenuate the effect of obfuscation, they interpret this state as having a strong causal importance. The difference between the corrupted output probability and the output probability after restoring one location is called the *indirect effect* of that location.

However, all of the analysis are based on observing individual causal traces. Meng et al. (2022) also proposed to compute the average indirect effect over larger quantity of sentences. In Table 13 we present our analysis of average indirect effects across different types of information. The result confirms our findings in Section 4.

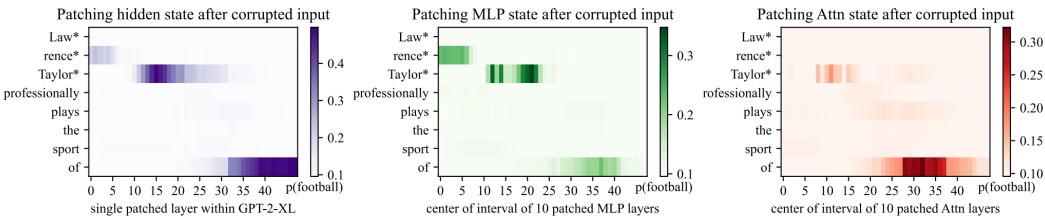

(a) Many factual causal traces also do not show this distinction. Example taken from Meng et al.'s (2022) Figure 10c. The MLP module show causality at both the early and the late site.

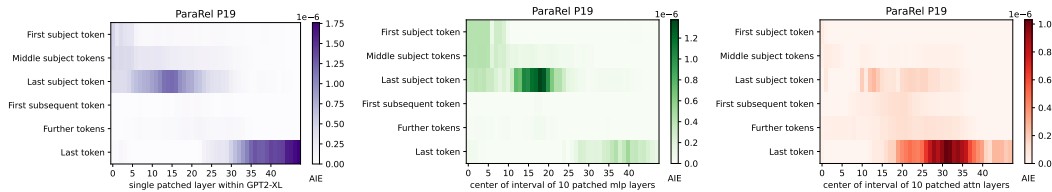

(b) We reproduced Meng et al.'s (2022) calculation of the average indirect effect of individual model components. We do see the separation between MLP and attention modules. However, we can also see that there is a weaker but discernible MLP causality at the late site. This shows that the previous example is not a negligible anomaly. The causal tracing pattern is less stable than Meng et al. (2022) originally conjectured.

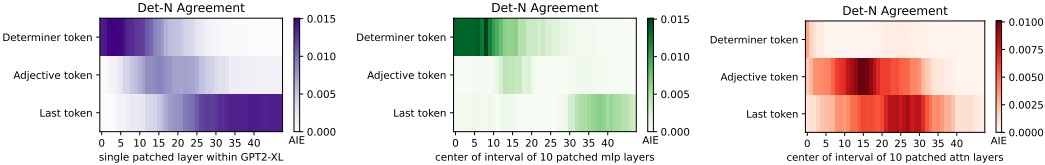

(c) The pattern that the MLP modules occupy an early site and attention modules occupy a late site do not persist for determiner-noun agreement. We used the determiner_noun_agreement_with_adjective_1 paradigm for this experiment as it contains the adjective token that is analogous to the first subsequent and further tokens that Meng et al. (2022) investigated.

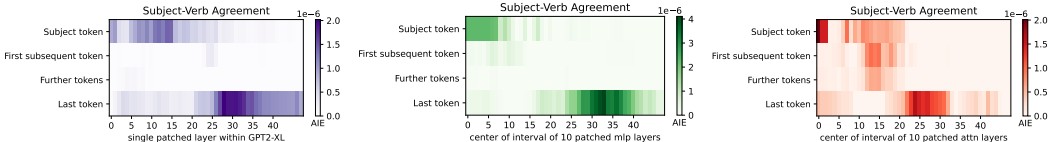

(d) The pattern of two distinct early and late sites is less apparent for subject-verb agreement. In fact, the MLP modules have the strongest causality at higher layers (30-35) than attention (~25).

Figure 13: Average indirect effect of different model component over multiple samples.

Table 10: The construction of our evaluation corpus for relation symmetry and synonym replacement

| Relation | | Edit Prompt | Evaluate Prompt |
|---|---|---|---|
| Symmetry P1376 | Template: Example: | [S] is the capital of [T→T*]. Ottawa is the capital of Canada→Italy. | The capital of [T*] is [S* →S]. The capital of Italy is Rome→Ottawa. |
| Symmetry P36 | Template: Example: | The capital of [S] is [T→T*]. The capital of Canada is Ottawa→Rome. | [T*] is the capital of [S* →S]. Rome is the capital of Italy→Canada. |
| Synonym P101 | Template: Example: | [S] works in the field of [T→T*]. Anaxagoras works in the field of philosophy→linguistics. | [S] is a [$T_s$ →$T_s^*$] Anaxagoras is a famous philosopher→linguist. |

# E  EVALUATION OF SYMMETRY AND SYNONYM

We construct the datasets used for both the symmetry and synonym evaluation from PARAREL relations. Table 10 shows an overview of the data construction process.

**Symmetry**   We used the two bijective relations: P1376 (capital of) and P36 (capital) to evaluate the symmetry property.

For each P1376 (capital of) relation $(s, t, r)$, we edit the model using the prompt "[S] is the capital of" and change the target from $t$ to $t^*$. $t^*$ is another city from the corpus. Then, we prompt the model with "The capital of [T*] is" and see if the model outputs $s$ with a higher probability than the original $s^*$. We identified 234 relations in total.

Similarly, for relation P36 (capital), we edit the model using the original prompt "The capital of [S] is" and try to change the target from $t$ to $t^*$. $t^*$ is another country/state from the corpus. Again, if the model outputs $s$ with a higher probability than the original $s^*$, we count the evaluation as success. We identified 703 relations.

**Synonym**   For synonym replacement, we use P101 (field of work). We first rewrite each field of work to the occupation name. For example, *linguistics → linguist* and *aviation → pilot*.

Through the process, we identified several mistakes in the original P101 data. For example, some of the field of work targets are already names of occupation, resulting in ill-formed prompts such as "Clyde Tombaugh works in the field of **astronomer**." Some of the fields of work, typically country names, cannot be converted into occupation names. For example "Mark Mazower works in the field of **Balkans**." We discard data entries with these issues and we collect 50 distinct field of works with their occupations. Finally, we obtain 568 entries.

We edit the model with the original prompt "[S] works in the filed of" and edit the target from $t$ to $t^*$. $t^*$ is another field of work we collected from P101. Then, we use the prompt "[S] is a" to elicit response from the edited model. Let $t_s$ and $t_s^*$ be the two occupation names correspond to $t$ and $t^*$, we want to see if the model can also assign a higher probability to the new target synonym $t_s^*$ than the original $t_s$.

For example, first we edit the model from "Anaxagoras works in the field of philosophy" to linguistics. Then, we prompt the model with "Anaxagoras is a famous" an see whether the model assign a higher probability to *linguist* rather than *philosopher*.

