# OpenReview forum: "What does the Knowledge Neuron Thesis Have to do with Knowledge?"
_ICLR.cc/2024/Conference — ICLR 2024 spotlight_

### Official Review · Reviewer_7T91 · 2023-10-30

**Soundness:** 4 excellent
**Presentation:** 2 fair
**Contribution:** 4 excellent
**Rating:** 8
**Confidence:** 3

**Summary:**

This work revisits the hypothesis that knowledge in pre-trained transformers may be limited to a few neurons. They do this by running two different model editing methods to identify neurons that are responsible for specific syntactic phenomena using the BLIMP minimal pairs dataset. While they do find a small number of mostly local knowledge neurons responsible for syntax, they find that intervening on these neurons is not enough to change model predictions in a robust way.

The paper also borrows from past work on factual editing, and along with their own results, conclude that the knowledge neuron thesis is flawed because intervening on these neurons is not sufficient to reliably change model behavior. Thus, they call for a more "holistic" approach that studies not individual neurons but entire layer structures and attention mechanisms as the motif for interpretability.

**Strengths:**

- The paper is extremely well-written with clear arguments, and experimentation.
- The results presented bring a lot of clarity to interpretability of transformers
- A lot of previous model editing techniques fail to systematically study if effect of edits are just local or if they are systematic, while this work does this very comprehensively.
- In my understanding, there is no prior work that applies editing techniques to identify "syntax neurons" and this aspect of the paper is quite novel

**Weaknesses:**

One weakness of the paper is that some of the presentation of experiments could be cleaned up substantially. Some specific suggestions for improvements:

- Results in 3.2 (first paragraph) seem quite loaded. This paragraph presents attribution scores, shows that identified neurons have regularity, the affect of causal interventions, how identified neurons have more to do with frequency cues than syntax etc. I think these results could be broken up into their own paragraphs.

- It would also be great to clearly show which results are taken from prior work. I suspect atleast Table-1 and Figure-5(c) is taken from prior work?

**Questions:**

- I would be interested to know what the authors think interpretability research should focus on i.e. it appears that knowledge is mostly distributed and not necessarily isolated to specific neurons. There is some discussion around using attention weights and the underlying model circuit towards the end but it would be good to have a slightly more extended discussion around this.

- Some missing references on "decision making circuits":
  - Clark et al. 2019 (What Does BERT Look at? An Analysis of BERT’s Attention): They use attention patterns to identify syntactic knowledge in models.
  - Murty et al. 2023 (Characterizing Intrinsic Compositionality in Transformers with Tree Projections): They identify tree-structured circuits as a way to study generalization in transformers.
  - Wu et al. 2023 (Interpretability at Scale: Identifying Causal Mechanisms in Alpaca): Finds alignments b/w model hidden states and symbolic algorithms.

---

> ### Author Response · Authors · 2023-11-15
>
> Thank you so much for the insightful review!
>
> We agree that the results presented in the first paragraph of section 3.2 are overly compact.  We have reflected that and we will break it into several paragraphs in the updated version of the paper.
>
> Table 1 presents our own results from our newly proposed criteria and datasets. Figure 6c was Yao et al.'s (2023) evaluation. We will include a citation in the figure caption as well to avoid the confusion.
>
> - **Q1: Future interpretability research directions.**
>
>     In our opinion, we think the recent work that tries to identify “circuits” in PLMs is a promising direction. Our paper shows that the recall of facts and the expression of linguistic phenomena may follow similar mechanisms but they are certainly much more complex than a simple key-value dictionary.
>
>     However, the community is still at a very early development stage in creating this circuit mode of interpretation. The circuit identification methods are ad hoc and can only be applied to a small set of tasks. In future work, we will try to formalize the circuit interpretation framework and apply it to more tasks and phenomena.
>
>     We will extend our discussion to this in the updated draft. And we would like to thank the reviewer for reminding us of the circuit interpretation references.
>
> - **Q2: Missing references.**
>
>     Thank you for also reminding us of the Clark reference. We will include it, as well as the circuit references in the updated version of the paper.

---

### Official Review · Reviewer_c8FP · 2023-11-01

**Soundness:** 3 good
**Presentation:** 3 good
**Contribution:** 4 excellent
**Rating:** 8
**Confidence:** 4

**Summary:**

This paper reassess the Knowledge Neuron Thesis in two ways, with syntatic minimal pairs and with generalizing to bijective relationships and synonyms. Theranalysis shows the limitation of current knowledge identification and editing, suggesting the need for more sophisticated understanding of inner mechanism of a language model.

**Strengths:**

1. This paper introduces many new practices for the rigorous study of knowledge neuron thesis, including using minimal pairs and t-test.
2. Broadening the definition of knowledge neural and connecting it to prior works in linguist phenomena.
3. Through and diverse analysis.

**Weaknesses:**

1. If I have to nitpick, section 4 felt a bit disjoint from the rest of the paper and is not fully fledged.

**Questions:**

n/a

---

> ### Author Response · Authors · 2023-11-15
>
> Thank you for your review and the recognition!
>
> Thank you for the feedback on the flow of the paper. We will reword the end of section 3 and section 4 to improve continuity.

---

### Official Review · Reviewer_ZPK4 · 2023-11-01

**Soundness:** 3 good
**Presentation:** 3 good
**Contribution:** 2 fair
**Rating:** 6
**Confidence:** 3

**Summary:**

This paper examines the “knowledge neuron” hypothesis in a number of pretrained LLMs, namely, that factual knowledge can be localized to a small number of neurons, and that ablation of those neurons alters the probability of, and/or the final chosen output token. It further extends knowledge to include syntactic or formal knowledge, and similarly finds small number of neurons that can be ablated to suppress their respective represented knowledge, particularly distributed throughout the later layers. However, through previous and additionally proposed metrics, in particular emphasizing bi-directionality and synonym-agnosticism, the authors argue that the discovered knowledge neurons cannot be considered to contain anything like “knowledge”, but simply conserve token correlations found in the training text.

**Strengths:**

Overall, I found the paper to be clearly written, except for some very technical and linguistics-specific concepts that warrant more explanation (and earlier). Its usage of intuitive examples and graphical illustrations throughout the text was very helpful for me to understand its arguments. Lastly, the experiments seem comprehensive, and convincingly demonstrates the authors’ two main claims: the existence of syntax-knowledge neurons in LLMs analogous to fact-knowledge neurons, and that neither sets of knowledge neurons can be considered to robustly represent “knowledge”.

**Weaknesses:**

Despite its technical soundness, I’m personally struggling to understand the significance of these findings on a larger scale, though I must admit that this is not my field. In particular, I feel that such detailed investigations of LLMs on a more “cognitive” level, i.e., assigning individual neurons to be representing concepts / knowledge wholesale, is orthogonal to dissecting the computational mechanisms of attention-based LLMs, and is more suitable for a conference like ACL. This is not really the fault of this particular paper, but the literature they attempt to address (which are predominantly published in ACL), though ironically, this paper raises exactly the point that such “knowledge neuron” search in LLMs may be ill-advised, given that these models are simply token sequence autocomplete machines.

Nevertheless, given its current scope and that we are explicitly asked to assess significance of contribution to the field (of machine learning), I recommend borderline rejection for ICLR (but would otherwise strongly recommend acceptance for, e.g., ACL!). But I would be willing to convinced that it’s within scope if the AC and other reviewers disagree.

**Questions:**

- one of my major concerns is the neuron selection procedure, which a priori limits the number of knowledge neurons to be 2-5. As I understand it, this procedure was not proposed by the authors, but in my opinion this process alone excludes the possibility of distributed representation, a much more reasonable null-hypothesis, resulting in a seemingly important discovery of knowledge neurons but in fact represents very little of the actual computations in the model. This is very reminiscent of the “grandmother” or “Jennifer Aniston” neuron type of work in neuroscience, and narrows the scope of the investigation arbitrarily and prematurely. Some investigation of distributed representation would, in my opinion, increase the impact and reach of this work

- the paper seems to be “on the fence” and sometimes explicitly contradicting itself. For example, page 7 states “LMs process and express the two types of knowledge using the same mechanism.” while the high-level conclusion of the paper, iiuc, is that there is no “knowledge neurons”. I think it would improve readability if the authors can find a more consistent messaging.

- the paper references a small number of previous works heavily, and without prior knowledge in the field, it’s hard for me to assess how much of the contributions are novel. The assessment of syntactic knowledge and newly proposed metrics are clearly new contributions, but a small “contributions” section explicitly and concisely summarizing this would be helpful.

- most of the illustrative examples (Figs 2-6) are on the case of determiner-noun, and some successful examples of the other two cases (subject-verb, gender and number agreement) would be even more convincing. Apologies if I had missed this in the supplemental.

- Formal and functional competence are referenced in the paragraph after figure 1, but without definitions, which can be confusing for a naive reader. The definition in the later paragraph was very helpful, and may be better if moved to be earlier.

---

> ### Author Response · Authors · 2023-11-15
>
> Thank you for the great review and the insightful feedback!
>
> However, we want to respectfully disagree with the reviewer's assertion that our work is studying LLMs on a "cognitive" level and not suitable for ICLR. Our paper is about interpreting and understanding the function of each transformer component to bring direct model editing of LMs to fruition. If the knowledge neuron thesis were to hold, direct model editing could be implemented by simply editing the MLP neuron activations or weights. This will significantly enhance the safety and privacy aspects of LLMs. But as we proved in this paper, such a simple fix does not exist. The limitations of current KN-based model editing methods originated from the limitation of the framework that only edits MLP neurons. Future improvement with only simple neuron manipulation may be a futile endeavour. We must resort to a more intricate means of interpretation and manipulation beyond the individual neuron level. Therefore, our work is about interpreting the representations within the transformer architecture learned by LLMs, which has more of a focus on the architecture of transformers than linguistics. This is also why we chose to submit to ICLR rather than an *ACL conference, despite similar deadlines.
>
> The misconception may have arisen from our citation of formal vs. functional competence work from cognitive science. In fact, in this respect, we are "anti-cognitive," because we dispute such appeals to pseudo-cognition in mechanistic interpretability work. In this paper, we disputed two such points in prior work, i.e., Dai et al.'s (2022) attribution of layer depth to the distinction between syntax and semantics, and both Dai et al.'s (2022) and Meng et al's (2022, 2023) strident use of the term "knowledge." In the paper we called for grounding the interpretation of the computational mechanisms of LLMs to robustly defined, human-interpretable tasks such as syntactic phenomena rather than broad analogies to human cognition.
>
> Lastly, we wish to point out that the ICLR references we have cited in our paper are very much in the same vein as our work, and so there shouldn't be a question as to the suitability of our paper to ICLR. Our work re-examines current mechanistic interpretability work (Meng et al., 2023) and, as we have discussed with reviewer @7T91, we concur with the future trend of interpreting LMs with "circuits" (Wang et al., 2022). So both sets of papers — the past and future of our research programme — were published at ICLR.
>
> - **Q1: Neuron selection procedure.**
>
>   The KN thesis hypothesizes that "knowledge" can be localised to a small number of neurons. There are two parts to this thesis: 1) the representation is localised to a small number of neurons, and 2) the representations that LMs capture encode "knowledge." Our paper confirms and extends the first part of the thesis to syntactic tasks, but refutes the second part — the representation localises complex patterns, but cannot, by any commonly accepted definition, constitute knowledge. We did not change the neuron number limit because the MLP weights can indeed still be interpreted as patterns in a localised fashion.
>
>   We also strongly agree with the reviewer's intuition that knowledge can reside in the network in a much more distributed fashion, and this is the message we are trying to communicate. We further emphasize this in the conclusion of the paper. It is obstinate and unreasonable to claim that LMs do not capture any knowledge at all, yet it is oversimplistic to adopt the view of the KN thesis to claim that knowledge is directly stored in the MLP weights just as in key-value dictionaries. Future breakthrough in mechanistic interpretability must adopt a more distributed approach that looks beyond the MLP weights. We are dedicated to moving towards that goal in our future work.
>
> - **Q2: Consistent messaging.**
>
>   As the reviewer has summarized, the main point of this paper is that "the existence of syntax-knowledge neurons in LLMs analogous to fact-knowledge neurons, and that neither sets of knowledge neurons can be considered to robustly represent 'knowledge'." This is a wording issue. We re-examined the paper and have updated the wordings to make it more consistent. For example, the sentence on page 7 is reworded as "LMs solve the two types of tasks (syntactic and factual) using the same mechanism."
>
> - **Q3: A "contributions" section.**
>
>   Such a good point! We will add such a section.
>
> - **Q4: Illustration of other linguistic phenomena.**
>
>   Because of space limitations, we must place the other linguistic phenomena (subject-verb, gender and number anaphor agreement) in the appendix.
>
> - **Q5: Definition of formal and functional competence.**
>
>   Thank you for the suggestion. This will certainly provide the readers with more context. We will move the brief definition of formal and functional competence to the introduction where it first appears.

---

### Meta-Review · Area_Chair_rFN2 · 2023-11-27

**Metareview:**

This paper looks at a popular approach for model interpretability and control and evaluates it extensively including its generalization and add more nuance (syntactic vs formal). It's well written. While relatively linguistics-y and possibly a bit niche, LLMs interpretability is a pretty central issue to most of ML right now (and this paper addresses a method actively in use), so should be represented at the conference.

**Justification For Why Not Higher Score:**

N/A

**Justification For Why Not Lower Score:**

Well written, extensive expts, relevant topic.
I'd be ok with this being a poster vs oral, but think it should be at the conference.
All reviewers recommend acceptance.

---

### Decision · Program_Chairs · 2024-01-16

Accept (spotlight)